# Wetting and complex remodeling of membranes by biomolecular condensates

Agustín Mangiarotti [1], Nannan Chen [1,2], Ziliang Zhao [1,3,4], Reinhard Lipowsky[1] & Rumiana Dimova [1] ✉

Cells compartmentalize parts of their interiors into liquid-like condensates, which can be reconstituted in vitro. Although these condensates interact with membrane-bound organelles, their potential for membrane remodeling and the underlying mechanisms of such interactions are not well-understood. Here, we demonstrate that interactions between protein condensates - including hollow ones, and membranes can lead to remarkable morphological transformations and provide a theoretical framework to describe them. Modulation of solution salinity or membrane composition drives the condensate-membrane system through two wetting transitions, from dewetting, through a broad regime of partial wetting, to complete wetting. When sufficient membrane area is available, fingering or ruffling of the condensate-membrane interface is observed, an intriguing phenomenon producing intricately curved structures. The observed morphologies are governed by the interplay of adhesion, membrane elasticity, and interfacial tension. Our results highlight the relevance of wetting in cell biology, and pave the way for the design of synthetic membrane-droplet based biomaterials and compartments with tunable properties.

The last decade of research has provided ample evidence that in addition to membrane-bound organelles, cells compartmentalize their interiors into membraneless organelles, also referred to as macromolecular condensates or coacervates, which behave as liquid droplets. Examples include nucleoli, Cajal bodies, P-bodies, and stress granules, all of which represent liquid protein/RNA-rich droplets within the cell[1–3]. They arise from the condensation of cellular material through liquid-liquid phase separation and can be reconstituted in vitro[4–6]. This discovery has expanded research on the functional roles of liquid droplets in animal, fungal, and plant systems, including compartmentalization, sorting of macromolecules, tuning of enzymatic reactions, preservation of cellular fitness, immune response, and temperature sensing[1,3,7]. In addition, aberrant protein condensation has been proposed as an intermediate step in neurodegenerative diseases such as Parkinson's and Alzheimer's[3,8]. Thus, research on

membraneless organelles has become a very active and strongly interdisciplinary field[7].

One interesting and little-explored aspect of liquid droplets is that although termed "membraneless", they can come into contact with and wet membranous compartments[9,10]. In the last years, it has been found that membrane-droplet interactions are involved in key biological processes, such as signal transduction in T cells[11], assembly of virus capsid proteins[12], biogenesis and fission of ribonucleoprotein granules in the endoplasmic reticulum[13,14], development of tight junctions[15], and assembly of endocytic vesicles[16]. Despite this progress, a detailed understanding of the underlying physicochemical mechanisms and systematic characterization of membrane-droplet interactions is still missing because these interactions are difficult to assess in vivo, in part due to the small size of condensates involved[17].

[1]Max Planck Institute of Colloids and Interfaces, Science Park Golm, 14476 Potsdam, Germany. [2]Present address: Department of Nutrition and Food Hygiene, Guangzhou Medical University, Guangzhou 511436, China. [3]Present address: Leibniz Institute of Photonic Technology e.V., Albert-Einstein-Straße 9, 07745 Jena, Germany. [4]Present address: Institute of Applied Optics and Biophysics, Friedrich-Schiller-University Jena, Max-Wien Platz 1, 07743 Jena, Germany. ✉e-mail: Rumiana.Dimova@mpikg.mpg.de

So far, membrane-droplet interactions have been studied in the context of only a few in vitro systems. When aqueous two-phase systems (ATPS) consisting of polymer solutions[18–22] were encapsulated or put in contact with giant unilamellar vesicles (GUVs)[23], several biologically relevant processes were demonstrated to occur: outward or inward budding, mimicking exo- and endocytic processes; formation of membrane nanotubes; and fission of vesicle compartments[22,24]. These phenomena were triggered either thermally or via osmotically-imposed concentration changes. Another approach involves inducing coacervation in the interior of GUVs by externally changing the pH[25]. The uptake of coacervate droplets by vesicles via endocytosis was demonstrated to be modulated by electrostatic droplet-membrane interactions[26]. Furthermore, membrane tubulation was observed in GUVs decorated with anchored proteins that underwent phase separation[27].

While several studies have already addressed the effect of membrane composition and phase state on interactions with condensates[27,28], as well as the coupling between lipid domains and condensates[29,30], we focused on understanding the mechanism of membrane wetting and remodeling by condensates. Here, we provide a systematic analysis of the membrane remodeling and wetting behavior of GUVs exposed to water-soluble proteins that phase separate into a protein-rich and a protein-poor phase. As a model protein, we employed glycinin, which is one of the most abundant storage proteins found in the soybean. Glycinin undergoes liquid-liquid phase separation in the presence of sodium chloride[6,31], making it a convenient model system to study membrane-droplet interactions. When combining glycinin condensates with GUVs we observed condensate-membrane adhesion, partial and complete wetting, condensate-induced budding, and a remarkable feature of this interaction: the complex remodeling of the membrane-droplet interface, resulting in ruffled membrane-condensate structures that resemble fingers. Furthermore, shifts within the phase-coexistence region of the glycinin phase diagram generate hollow condensates capable of spreading on membranes, thus expanding the possibilities for cell-mimetic compartmentalization. These diverse phenomena and morphologies are summarized in Fig. 1. We demonstrate that our findings do not only apply to glycinin condensates, but also to peptide coacervates and PEG/dextran ATPS. Our study has important implications for the droplet-induced morphogenesis of ubiquitous membrane-bound organelles. It also has applications for material science and synthetic biology in the development of smart soft materials with compartmentalization and wetting-modulated morphology.

## Results

### Wetting transitions of biomolecular condensates on membranes

Liquid droplets at surfaces adopt different shapes depending on the strength of interaction: they may remain spherical (no wetting); spread slightly on the surface, adopting the morphology of a truncated sphere (partial wetting); or spread completely, wetting the whole surface (complete wetting). By changing control parameters such as surface or droplet composition, temperature, or interaction strength, the system can undergo a transition from dewetted to partial wetting and even

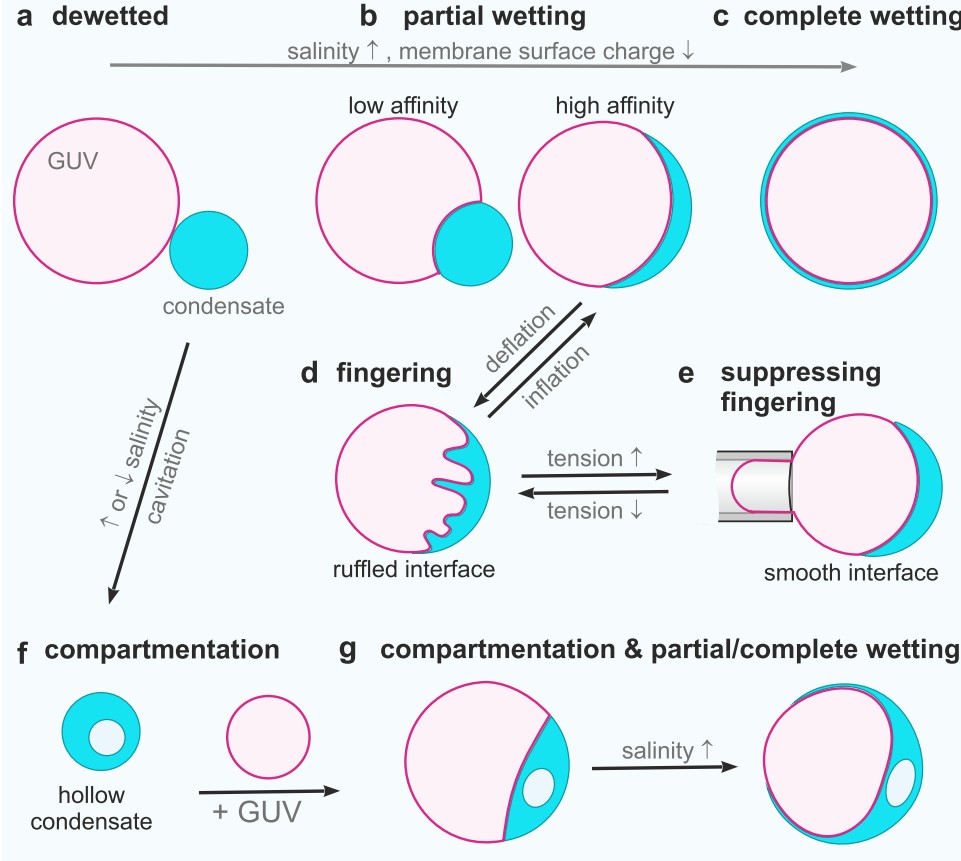

**Fig. 1 | Sketch illustrating wetting transitions and observed morphologies of the membrane-condensate system.** Membranes (magenta) in contact with biomolecular condensates (cyan) can undergo two wetting transitions from dewetting (**a**, **f**) through partial wetting (**b**, **g**-left) to complete wetting (**c**, **g**-right) modulated by ionic strength and membrane composition. Complex morphological transformations exhibited by interface ruffling and fingering (**d**) can be observed when excess membrane area is available, or suppressed when this area is retracted upon tension increase (**e**). Salinity changes can result in the formation of hollow condensates (**f**), which can also exhibit partial and complete wetting (**g**) offering additional means of compartmentation in cells.

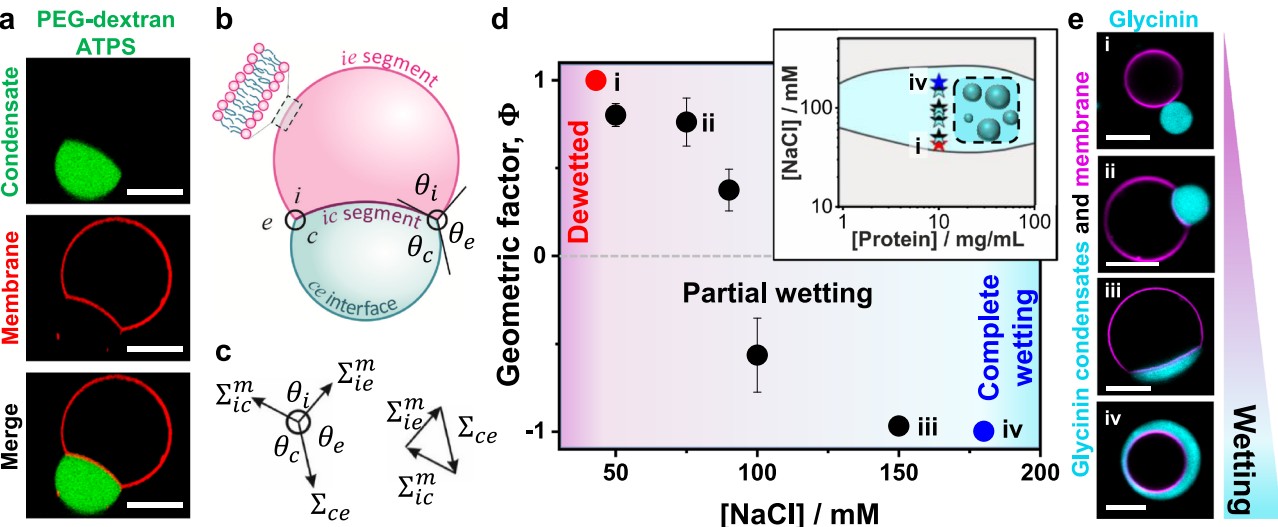

**Fig. 2 | Membrane wetting by PEG/dextran and glycinin condensates.**
**a** Confocal microscopy images showing partial out-wetting of a vesicle (red) by a dextran-rich droplet (green) in phase-separated PEG/dextran aqueous two-phase systems; the vesicle adopts a moon-like shape. **b** Partial wetting leads to the formation of a contact line between the $ce$ interface (blue line) and the membrane. The contact line partitions the membrane into the $ie$ (magenta) and $ic$ (purple) segments, with the contact angles $\theta_c + \theta_i + \theta_e = 360°$. **c** Force balance between the droplet interfacial tension $\Sigma_{ce}$ and the mechanical tensions $\Sigma_{ie}^m$ and $\Sigma_{ic}^m$ within the two membrane segments. These three tensions form the sides of a triangle (which implies Eqs. 9 and 10 in "Methods")[52]. **d** Experimental data for the geometric factor $\Phi = (\sin\theta_e - \sin\theta_c)/\sin\theta_i$ (Eqs. (8)–(11) in "Methods") as a function of NaCl

concentration. The largest possible value $\Phi = +1$ (red circle) corresponds to the transition from dewetting to partial wetting at [NaCl]≈43 mM, the smallest possible value $\Phi = −1$ (blue circle) reflects the transition from partial wetting to complete wetting at [NaCl]≈180 mM. All data: mean ± SD, $n = 10$ per composition. Data for the geometric factor of the PEG/Dextran system can be found in Supplementary Fig. 1b. The inset shows the phase diagram of glycinin in salt solutions with the region of phase coexistence shown in cyan. The locations of the measured geometric factor data points are indicated with stars. **e** Examples of confocal microscopy images of different wetting morphologies for the data points indicated in (**d**); for more images see Supplementary Fig. 1a. All scale bars: 10 μm. Data for panel d are provided as a Source Data file.

complete wetting[32,33]. Similarly, when biomolecular condensates come into contact with membranes, they can undergo wetting transitions depending on the interaction strength[19]. Contrary to solid substrates, however, membranes can deform because of their relatively low bending rigidity. Membrane wetting transitions were first described for vesicles in contact with ATPS composed of mixtures of poly(ethylene glycol) (PEG) and dextran[19]. In these systems, several morphological transformations have been produced by modulating the wetting of the membrane by the polymers[21,34]. Figure 2a shows an example of partial wetting morphology for a vesicle in an ATPS system. An out-wetting morphology can be observed where the condensate droplet (green) is from the dextran-rich phase. Similar examples for glycinin condensates in contact with membranes are shown in Fig. 2e. To quantify the membrane-condensate interactions, we first provide a theoretical description of the system (see Methods for details): in general, partial wetting morphologies involve three different surface segments, which meet along a contact line (Fig. 2b), with three corresponding contact angles and surface tensions (Fig. 2c)[34–37]. The contact angles and the mechanical tensions, $\Sigma_{ic}^m$ and $\Sigma_{ie}^m$ of the two membrane segments depend on the lateral stress, $\Sigma$ within the membrane ("Methods", Eq. 8), and thus on the size and shape of the vesicle. In contrast, both the interfacial tension, $\Sigma_{ce}$, and the affinity contrast, $W$ ("Methods"), Eq. (9), which represents the difference between the adhesion free energies per unit area of condensate and external buffer, are material parameters that do not depend on the size and shape of the chosen condensate-vesicle couple. Furthermore, the ratio of the affinity contrast, $W$, to the interfacial tension, $\Sigma_{ce}$, is directly related to the three contact angles and can thus be obtained by measuring these angles from microscopy images ("Methods", Eq. (11), see also Fig. 3b). Using this description of geometry and membrane elasticity, we systematically studied the interaction of GUVs made of zwitterionic phosphatidylcholine with glycinin condensates[6] and of GUVs enclosing PEG-dextran ATPS. Upon interaction of the condensates with the membranes, the system undergoes two wetting transitions at constant

temperature (see Fig. 2d, e, Supplementary Fig. 1a, c, d for data on glycinin and varied membrane composition; analogous trend is observed upon raising ATPS polymer concentrations, see Supplementary Fig. 1b). At low salt content, the glycinin droplets do not interact with the vesicles (dewetted state, $\Phi=1$). Raising the salinity leads to increased attractive interaction between the droplets and the membrane (partial wetting, $-1<\Phi<1$) until complete wetting ($\Phi=-1$), i.e., spreading of the condensates over the whole membrane surface, which is reached when the salinity exceeds 180 mM NaCl (Fig. 2d, e, Supplementary Fig. 1a,d). The observed behavior plausibly results from charge screening of the protein droplets (glycinin is negatively charged at the working pH=7), allowing for interaction with the membrane. The experimental data show that glycinin condensates interacting with GUVs exhibit three different wetting regimes separated by two wetting transitions at a constant temperature. Even though wetting transitions have been extensively studied in the past[33,38,39], to the best of our knowledge, this behavior has not been previously reported, ascribing remarkable features to this condensate-membrane system.

The wetting transitions can easily be tuned, not only by changes in salinity, but also by modifying the membrane composition (Fig. 3, Supplementary Fig. 2). When increasing the absolute membrane charge by including negatively (DOPS) or positively charged lipids (DOTAP), the system moves towards dewetting (Fig. 3a, b). This behavior is counterintuitive, as we expected that increasing the positive charge in the membrane would lead to a stronger interaction with the negatively charged protein, but the opposite is observed. Affinity of the condensates for both negatively and positively charged membranes is decreased compared to neutral membranes, suggesting that their interaction with the membrane is not only electrostatic in nature, but presumably also hydrophobic. Previous reports have described favorable hydrophobic interactions between hydrophobic glycinin residues and phosphatidylcholines or lecithin[40,41]. It is important to note that glycinin is a relatively large molecule (a hexamer, roughly

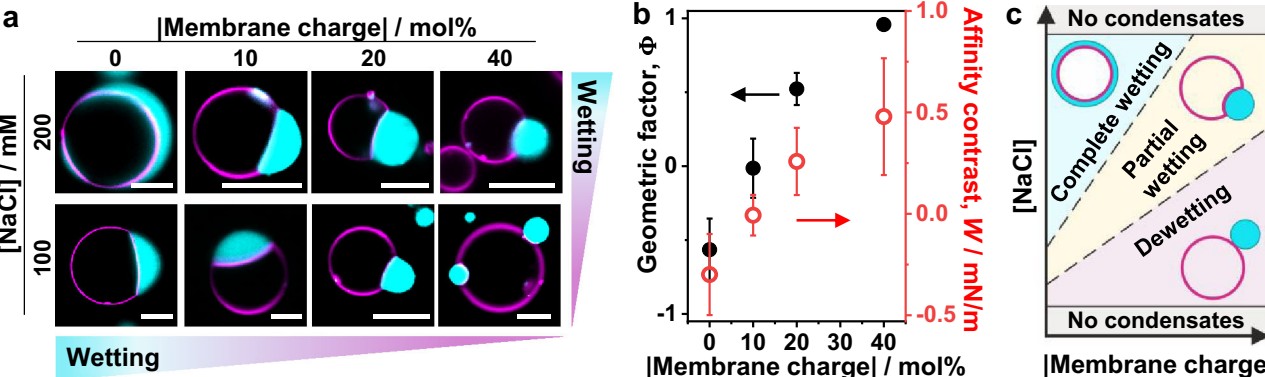

**Fig. 3 | Tuning wetting by membrane charge and salinity. a** Confocal microscopy images of the vesicle-droplet system showing characteristic wetting morphologies as a function of salinity and membrane composition (in terms of absolute value of the membrane charge), see Supplementary Fig. 2 for more data. Scale bars: 10 μm. **b** Geometric factor Φ (black axis) and affinity contrast $W$ (red axis) vs absolute membrane charge (% mol) for giant vesicles in contact with glycinin condensates at 100 mM NaCl (data for 10 mol% DOPS and 10 mol% DOTAP are combined, see Supplementary Fig. 2). The affinity contrast is given in units of interfacial tension ($W = \Phi\Sigma_{ce}$, see "Methods") and the errors are estimated with error propagation analysis. All data: mean ± SD, $n = 10$ per composition. **c** Schematic morphology diagram of the droplet-vesicle interactions as a function of salinity and membrane charge. Data for (**b**) are provided as a Source Data file.

5 nm in size), presenting acidic and basic residues that can undergo conformational changes upon phase separation[6,42]. Altogether, this suggests the involvement of hydrophobic interactions of glycinin with membranes, which are enhanced at higher fractions of neutral lipids. With increasing salinity, membrane charge is screened (Fig. 3a and Supplementary Fig. 2), favoring wetting. In this manner, membrane composition and salinity are simple parameters that allow for the fine-tuning of wetting morphologies and condensate stability (Fig. 3c).

Inspection of the partial wetting morphologies in Figs. 2 and 3 shows that the visible membrane segments, *ie* and *ic* are separated by an apparent kink in the membrane shape (see for example, the cusp of the moon-like vesicle seen in the membrane channel in Fig. 2a). The fine structure of this kink corresponds to a highly curved membrane segment that cannot be resolved with confocal microscopy, but can be visualized with STED microscopy[43]. This arises from the mechanical balance between the capillary forces, generated by the interfacial tension, $\Sigma_{ce}$ of the condensate-buffer interface and the membrane bending moment, which is proportional to its bending rigidity, $\kappa$. This mechanical balance implies that the contact line segment acquires a curvature radius on the order of $\sqrt{\kappa/\Sigma_{ce}}$. For typical values of bending rigidity, $\kappa \simeq 10^{-19}$ J and interfacial tension, $\Sigma_{ce} \simeq 0.5$ mN/m (measured here, see Methods), we obtain the curvature radius, $\sqrt{\kappa/\Sigma_{ce}} \simeq 14$ nm. This is indeed below the optical resolution limit and thus appears as a kink in the confocal images. The length scale, $\sqrt{\kappa/\Sigma_{ce}}$ enters the normal (or perpendicular) component of the force balance between the three surface tensions at the contact line as shown by minimization of the combined curvature and adhesion energy of the vesicle-droplet couple[35]. In the context of liquid droplets at the surfaces of solid materials, the competition between the surface tension, $\Sigma_{int}$ of the liquid-vapor interface and the Young's modulus, Y of the solid material leads to the elastocapillary length, $\Sigma_{int}/Y$[44,45]. Note that the elastocapillary length grows with increasing surface tension, whereas the curvature radius, $\sqrt{\kappa/\Sigma_{ce}}$ decreases with increasing interfacial tension. For droplets at thin solid plates, the competition between the plate bending modulus, $B$ and the droplet interfacial tension, $\Sigma_{int}$ leads to the length scale, $\sqrt{B/\Sigma_{ce}}$ which provides another definition of an "elastocapillary length"[46]. The latter definition is analogous to the length scale, $\sqrt{\kappa/\Sigma_{ce}}$, considered here in the context of fluid membranes.

## Droplet spreading and dynamics
In all cases, membrane wetting by the protein condensates was characterized by slow dynamics on the order of minutes to hours (Fig. 4a, Supplementary Movie 1). Once in contact, the angles characterizing

the condensate-membrane system required 20–30 min to reach equilibrium (Fig. 4b). This is most likely related to the high molecular weight of glycinin (360 kDa), which translates to the high viscosity of the condensates (~4.8 kPa s, see Methods), which is roughly three orders of magnitude higher than that of condensates formed by other proteins like FUS (0.7 Pa.s.) or PGL-1 (1 Pa.s.)[47]. The data in Figs. 2 and 3, as well as the results described in the following correspond always to the final equilibrated wetting morphology.

Compared to bare, condensate-free membranes (segment *ie* in Fig. 2b), vesicle segments in contact with condensates (segment *ic*) are characterized by lower fluidity as a result of protein-lipid interactions, with the immobile fraction in such membrane segments being (16 ± 5)%, as measured by FRAP (Fig. 4c, d). This indicates that the condensates not only induce membrane morphological transformations, but also impose dynamic constraints on the wetted lipid molecules.

## Hollow condensates and droplet-vesicle bridging provide a means for compartmentalization
Similar to the behavior observed for PEG-dextran ATPS in contact with membranes, glycinin condensate-vesicle interactions can lead to complete droplet engulfment as long as sufficient membrane area is available. A vesicle can bridge several condensates and condensates can engulf or bridge several GUVs. The necessary conditions for engulfment are partial or complete wetting and excess vesicle area (compared to the surface area of a sphere enclosing the same volume), sufficient to wrap around the condensate. Thus, wetting can give rise to complex condensate-vesicle architectures consisting of multiple compartments (Fig. 5, Supplementary Fig. 3), which demonstrates the role of capillary forces in membrane remodeling[10,37]. Similarly to what has been reported for coacervates[26], the engulfment of condensates by the vesicle membrane can separate individual droplets from each other and prevent their coalescence. This process changes the area of the interface between the condensate and the liquid bulk phase, thereby regulating the diffusive exchange of molecules between the droplets and the bulk phase. Hollow condensates are obtained by inducing phase separation within the protein-rich droplets through shifting the salinity within the coexistence region of the phase diagram[6] (see Fig. 5b,c and Methods). These structures can also reshape membranes in a similar way (Fig. 5d), thereby providing additional means of compartmentalization. Considering the abundance of cellular membranes in the cell interior, we speculate that bridging, separation, and further compartmentalization mediated by

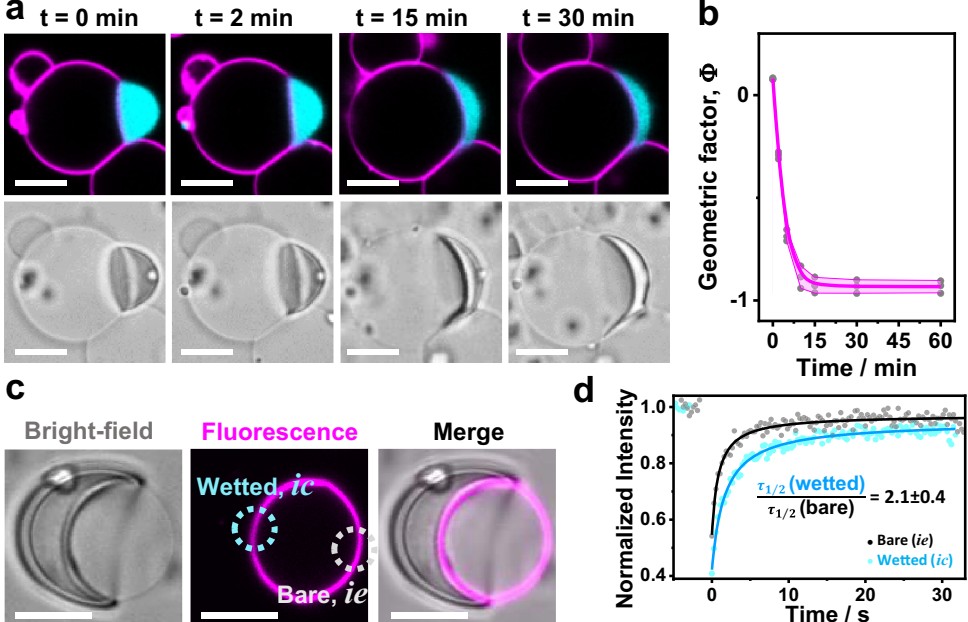

**Fig. 4 | Wetting and membrane dynamics. a** After the glycinin condensate (cyan) gets in contact with the membrane (magenta) at t = 0 min, the contact angles change slowly until reaching the final morphology, which remains stable for hours; confocal (top) and bright-field (bottom) images of the same vesicle-condensate pair. **b** Example of the measured geometric factor Φ for the membrane-condensate system shown in (**a**); DOPC membrane, 100 mM NaCl. The geometric factor was calculated using $n = 3$ images for each time, individual points are plotted (gray circles), the mean is indicated as a pink line and the SD is shown as a pink shadowed area. **c** Fluorescence recovery after photo-bleaching (FRAP) experiments show

decreased fluidity in the membrane segment that is wetted by the condensate (*ic* segment in Fig. 2b) compared to the bare one (condensate-free, *ie*) on the same vesicle. For these experiments, only the membrane was fluorescently labeled to avoid interferences from condensate fluorescence. The dotted circles shown in cyan/gray indicate the bleached regions in the *ic/ie* membrane segments respectively. **d** FRAP intensity curves yield halftimes of recovery $\tau_{1/2}$ which show that condensate wetting slows lipid diffusion by a factor of about 2 ($n = 5$); see "Methods". Scale bars: 10 µm. Data for panels b and d are provided as a Source Data file.

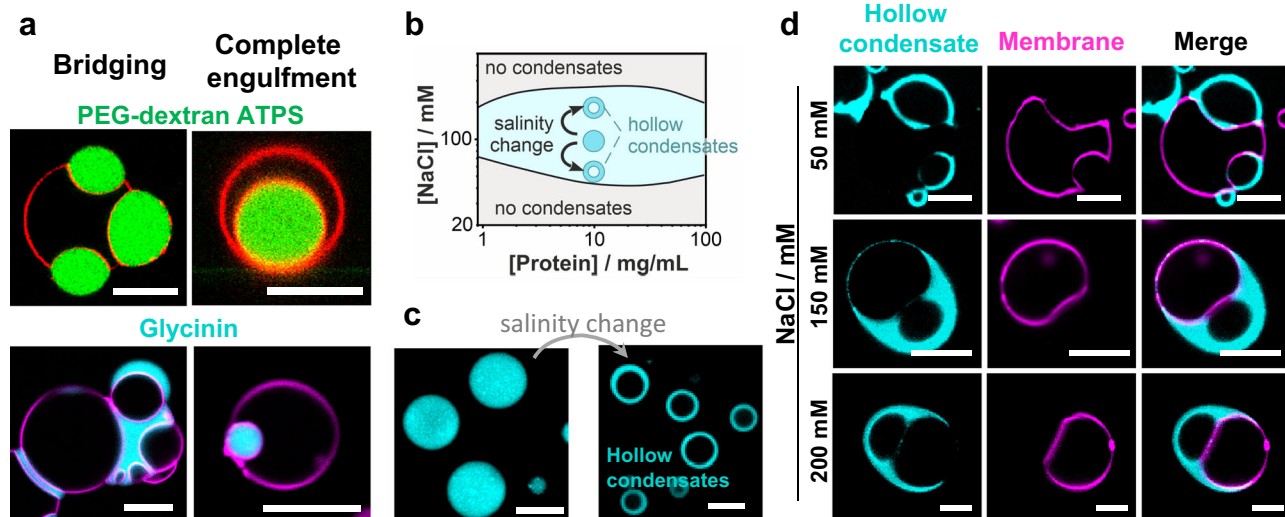

**Fig. 5 | Complex architectures and compartmentation generated by condensate-membrane interactions. a** Bridging and complete engulfment in condensate-membrane systems, where the condensate droplets are either dextran-rich (green) in PEG-dextran ATPS (upper row) or glycinin condensates (lower row). Either the vesicle membrane can bridge several droplets (top, left) or a condensate can bridge several vesicles (bottom left) creating complex architectures with several compartments. The engulfment of the condensate (right images) is energetically favored by both the increased contact area with the membrane and the reduced surface area of the *ce* interface (see Fig. 2b). **b**–**d** Hollow condensates

formation and membrane wetting. **b** Schematic illustration of the pathways in the glycinin phase diagram used to generate hollow condensates starting from droplets prepared in 100 mM NaCl and exposed to salinity shift to higher or lower concentration of NaCl, which triggers phase separation within the droplets. **c** Images of the condensates before and after the salinity change; here, from 100 mM NaCl (left) to 50 mM NaCl (right), showing the formation of hollow condensates by such a salinity downward shift. **d** Hollow condensates (cyan) in contact with soy-PC GUVs (magenta) at the indicated NaCl concentrations allow for additional system compartmentation (see also Supplementary Fig. 3). All scale bars are 10 µm.

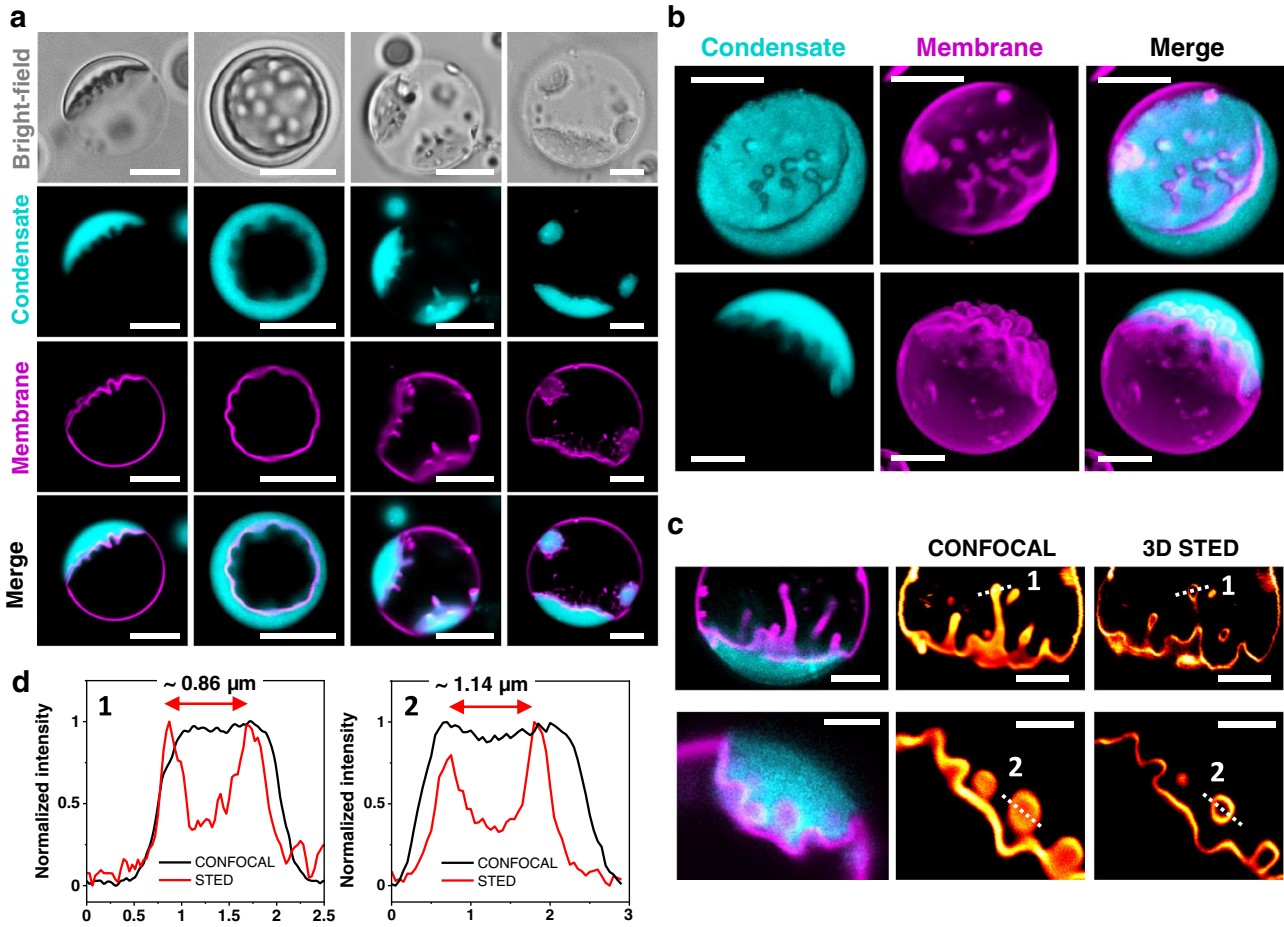

**Fig. 6 | Ruffling and fingering of the membrane-condensate interface leads to complex morphologies. a** Examples of ruffling morphologies of the condensate-membrane interface seen in bright field and confocal cross-sections. **b** 3D reconstructions of GUVs in contact with glycinin condensates showing the different ruffled morphologies (see also Supplementary Fig. 4). **c** 3D STED imaging resolves characteristic features of the fingered regions in the submicrometer range (see also Supplementary Fig. 5). **d** Intensity line profiles along the indicated dashed lines in the micrographs in (**c**) allow for an estimation of the finger thickness (left corresponds to the first image and right to the second one as indicated by the numbers in (**c**)). All scale bars: 5 μm.

membranes are highly relevant for the formation and organization of subcellular structures.

### Ruffling and fingering of the condensate-membrane interface

The most dramatic response to wetting in the glycinin-membrane system is observed in vesicles with excess membrane area. The membrane-condensate interface (*ic* segment in Fig. 2b) undergoes ruffling, which can proceed to excessive curving of the membrane/droplet interface (Fig. 6). This process of interfacial ruffling is reminiscent of "viscous fingering" observed when a low viscosity liquid displaces a higher viscosity liquid in a porous medium[48]. Our system differs in that a membrane separates the fluids and there is no porous confinement, nor fluid injection. A more relevant analogy is provided by the structural resemblance to elements of the endoplasmic reticulum, despite the missing network-like character and slightly higher curvature. The former could be created by subsequent fusion of the protrusions, e.g. by fusion proteins and the latter could be imposed by adsorbing or scaffolding proteins such as reticulons and atlastins.

The complex membrane shapes observed include large protrusions and invaginations in the form of finger-like structures and thin tubes (Fig. 6a–c, Supplementary Movie 2). The formation of these structures involves the mutual scaffolding of membrane and condensates, since both become curved (Fig. 6a, b, Supplementary Fig. 4a)

and the behavior is also observed for more complex membrane compositions (Supplementary Fig. 4b). The phenomenon is different from membrane tubulation triggered by spontaneous curvature generation, as observed in membrane wetting by ATPS, where tubes were observed to protrude always towards the PEG-rich phase (see e.g.[21,43,49].). Similarly, tubulation via compressive stress induced by protein phase separation at the membrane surface[27] can also be excluded because of the lack of directionality in the protrusions observed here. We observe that the interface can develop fingers pointing towards either side, presumably depending on the volume of the droplet or vesicle provided for the protrusions. Using 3D STED microscopy, we resolved characteristic features and dimensions of the reticulated (elastically deformed) regions, which lie in the submicron-micron range (Fig. 6c, d, Supplementary Fig. 5).

The formation of finger-like structures is slow. Ruffling begins while the contact angle is stabilizing (see Fig. 7a,b, Supplementary Movie 3) and the structures stabilize within 40–60 min. Vesicles exhibiting membrane ruffles or fingers were not considered when assessing the geometric factor, Φ in Figs. 2–4, because in such systems the contact angles are ill-defined. Once established, the ruffled structures do not fluctuate (Supplementary Movie 4) and can remain stable for hours. As shown in Fig. 7a (and Supplementary Movie 3), the contact area between the membrane and the condensate increases with the degree of ruffling.

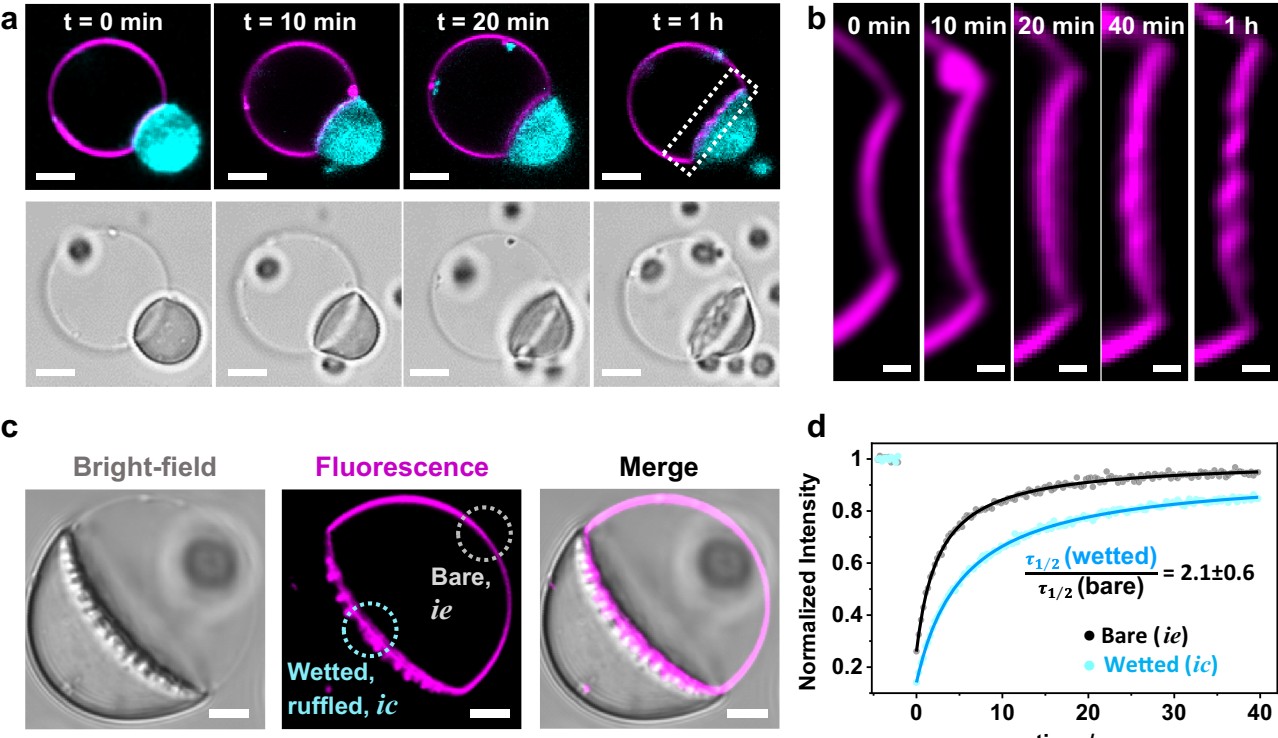

**Fig. 7 | Dynamics of membrane ruffling driven by glycinin condensates.**
**a** Mutual membrane-condensate molding and generation of curved structures at the membrane-droplet interface during wetting proceeds slowly (see also Supplementary Movie 3). **b** Zoom of the region indicated with a dotted line in (**a**). **c** Images in bright field and confocal cross section showing the two membrane segments examined by FRAP, the bare one (*ie* segment in Fig. 2b) and the wetted, reticulated one (*ic*). Only the membrane was labeled and it was bleached in the indicated regions. **d** Recovery curves for the reticulated region (cyan) and the bare condensate-free membrane (gray). From the fitting, the halftimes of recovery $\tau_{1/2}$ are obtained ($n = 5$, see "Methods"). Lipid diffusion in the membrane in contact with the condensate is twice as slow than in the condensate-free segment similarly to measurements on smooth interfaces where the slowdown is the same (compare to Fig. 4c, d). These results indicate that membrane ruffling by condensates does not alter lipid diffusion compared to that in wetted membranes where ruffling is absent. Scale bars in (**a**) and (**c**): 5 μm, in (**b**): 1 μm. Data for (**d**) is provided as a Source Data file.

The intricate and striking membrane morphologies observed arise from the gain of adhesion energy between the droplet and the membrane when excess area is pulled out of the *ie* segment and added to the *ic* segment (see Methods). Subsequently, the tension of this segment becomes negligible, $\Sigma_{ic}^m \approx 0$, which implies, via the force balance triangle (Fig. 2c), that $\Sigma_{ic}^m/\Sigma_{ce} \equiv \sin\theta_e/\sin\theta_i \approx 0$, and that the angle, $\theta_e$ should approach 180°. This is indeed consistent with our experimental observations, which show that the *ic* membrane segment becomes parallel to the *ce* interface along the contact line, i.e. the vesicle-condensate pair rounds up overall (see e.g. Figs. 6a, 7c). Furthermore, this tension relation together with Eqs. (10) and (11) imply negative values for the geometric factor, $\Phi$, and indeed, interface ruffling was observed only for [NaCl] ≥ 100 mM, where $\Phi < 0$, i.e. for negative affinity contrast. Whether or not the formation of finger-like structures is associated with an increase in the bending energy depends on the spontaneous curvature of the *ic* segment. The adhesion of the glycinin-rich droplet implies adsorption of the protein to the membrane and we would expect the membrane to bulge towards the droplet, as is the case for $\Phi < 0$. In this case, adhesion and bending act in a synergistic manner. At their liquid-liquid interfaces, condensates exhibit some degree of molecular organization, as shown previously by birefringence[6,50]. We cannot exclude that this surface organization is perturbed by the interaction of the membrane with the proteins at the droplet interface. However, the lipid mobility in the reticulated membrane segment remains similar to that in smooth membrane-condensate segments (compare Figs. 7c, d and 4c, d). We also confirmed that the membrane integrity is preserved during

wetting and ruffling (Supplementary Fig. 6). Furthermore, considering that solution asymmetry (in terms of salinity) across the membrane, as is the case for the GUVs examined here, can drive tubulation[51], we explored whether the interfacial ruffling would be preserved when exposing the membrane to symmetric salt conditions. We confirmed that ruffling is independent of asymmetric buffer conditions (Supplementary Fig. 7).

The interface ruffling and finger-like structures are only present in vesicles with available excess membrane area. To further study this aspect, we explored vesicle responses to osmotic deflation and inflation (see details in Methods). When vesicles are deflated by increasing the external osmolarity, they gain excess membrane area, as compared to the surface area of a sphere with equivalent volume. These vesicles can either exhibit visible fluctuations as quasi-spherical or floppy vesicles, or deform into non-spherical shapes where the excess area can be stored in the form of buds or nanotubes stabilized by spontaneous curvature[52,53]. In this context, interfacial ruffling becomes more pronounced upon stronger osmotic deflation (Fig. 8a). When the external osmolarity is decreased, vesicles become tense and spherical, thus suppressing fluctuations. This effect can be achieved with micropipette aspiration of the membrane ("Methods"), which increases membrane tension and suppresses ruffling, leading to smooth vesicle-droplet interfaces (Fig. 8b, Supplementary Movie 5). Subsequently decreasing the membrane tension by reducing the suction pressure restores ruffling. The tensions (~1 mN/m) required to smooth the interfacial ruffles are much higher than those needed to retract tubes formed in ATPS-vesicle systems (~20-100 μN/m)[49]. This

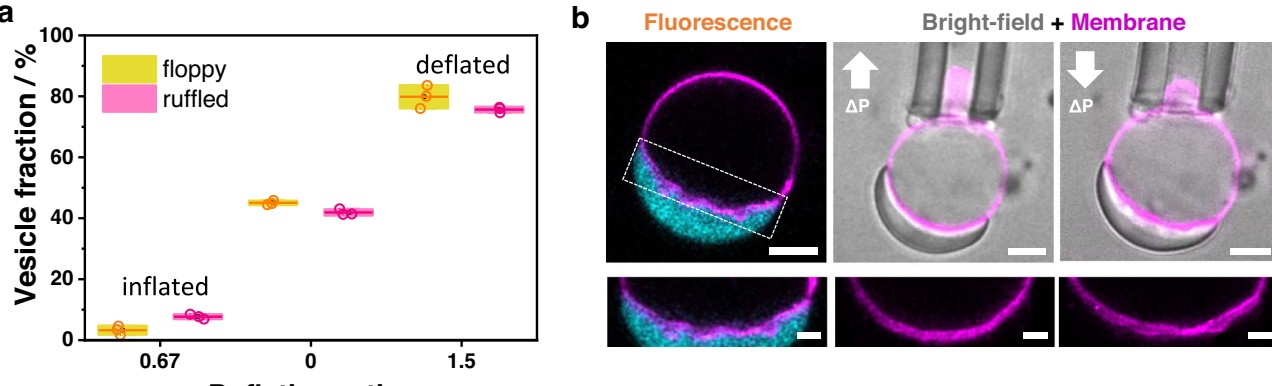

**Fig. 8 | Ruffling of the membrane-condensate interface is enhanced for vesicles with more excess area and can be tuned by tension. a** The percentage of vesicles in which ruffling is observed increases with increasing the membrane excess area as shown by the fraction of floppy vesicles. Deflation ratio in the graph is the ratio of the externally applied osmolarity to the initial vesicle osmolarity (n ≥ 50 for each experiment, three independent experiments are shown for each condition (circles), horizontal lines indicate the mean values and boxes width show the SD). **b** Increasing membrane tension via micropipette aspiration results in the suppression of ruffles and smoothing out of the condensate-membrane interface. The tension threshold needed to suppress the ruffles is (1.0 ± 0.2) mN/m (*n* = 5). When releasing the membrane area by decreasing the tension, ruffles reappeared within a minute (upper middle and right panels, up- and down-arrows signify increasing and decreasing suction pressure). Scale bars are 5 μm. A zoom of the membrane region highlighted with white dashed line is shown in the lower panels. Scale bars are 2 μm. Data for panel a is provided as a Source Data file.

difference can be attributed to the much higher interfacial tension of the protein condensates (-0.5 mN/m, Methods) compared to that of the ATPS (-0.01 mN/m)[54].

The above results on membrane remodeling and wetting transitions were obtained using glycinin droplets, but biomolecular condensates can exhibit a wide range of possible surface charges, interfacial tensions and viscosities[47]. It is important to note that the state of wetting of a condensate on a membrane may not only depend on the material properties of the condensate, but also on specific interactions between particular lipids or membrane components and condensate species. For glycinin droplets, the inclusion of charges in the membrane promotes dewetting, but this may not be the case for different kinds of condensates. This is exemplified by Dengue and Zika virus capsid proteins, which form condensates with DNA/RNA and wet negatively charged membranes[12]. To further study this, we performed experiments showing wetting transitions with condensate droplets composed of a polyamine, polydiallyldimethylammonium (PDDA) and adenosine triphosphate (ATP), as well as with condensates of the oligopeptide pair, poly-L-lysine ($K_{10}$) and poly-L-aspartic acid ($D_{10}$; see Supplementary Figs. 8 and 9). All three wetting regimes we observed in the glycinin system can be observed in these other systems, suggesting a general wetting mechanism for condensates on membranes. The wetting transitions in these highly charged systems (positive for PDDA/ATP droplets[26] and negative for $K_{10}/D_{10}$ condensates[55]) can also be modulated by membrane composition and salinity, as demonstrated for glycinin, albeit under different conditions (compare Fig. 3 and Supplementary Figs. 8 and 9). These results indicate a universal behavior and show that depending on the chemical nature of the condensates, different parameters need to be tuned to produce wetting transitions.

## Discussion

Our results demonstrate a wide spectrum of morphology changes driven by protein condensates interacting with membranes. Glycinin condensates in contact with GUVs were observed to undergo two distinct wetting transitions, with a broad intermediate regime of partial wetting (Fig. 2). We developed a theoretical framework that allows quantification of the membrane-condensate interaction and description of the system geometry by directly measuring the apparent contact angles with optical microscopy. The observed wetting behavior

can easily be modulated by salinity or membrane charge, as demonstrated with two additional condensate systems. Wetting transitions were also observed for a more complex lipid mixture containing cholesterol as a closer mimetic to biological membranes (Supplementary Fig. 1c). In this case, the values of the geometric factor are only slightly shifted towards higher NaCl concentration (see Supplementary Fig. 1d) compared to those for pure DOPC membranes shown in Fig. 2. This indicates that for this particular condensate-membrane system, increasing the membrane complexity and including a fast-flipping molecule such as cholesterol only has a minor effect on the wetting behavior compared to increasing the net surface charge of the membrane or changing the salinity.

Glycinin is a major storage protein in the soybean and plays a key role in plant embryogenesis. In most cotyledons, storage proteins are accumulated in the vacuoles[56], and it has recently been reported that they not only provide nutrients for subsequent growth, but also contribute to tonoplast, or vacuolar membrane remodeling during plant development[57]. Storage proteins form phase-separated droplets in the vacuole interior that interact with membranes, representing different wetting states, as observed in electron microscopy images[58,59] (see Supplementary Fig. 10a–e) and with fluorescence confocal microscopy[57] (Supplementary Fig. 10f). In the final developmental stage, the protein droplets undergo complete engulfment by the vacuolar membrane, forming protein bodies[58] (Supplementary Fig. 10c). Such remodeling processes are relatively slow, occurring over several days as demonstrated for protein storage vacuoles in *Arabidopsis thaliana*[57]. This is presumably related to the high viscosity of storage protein droplets, similar to what we report here for glycinin. The precise mechanisms by which these processes occur in plant cells remains unknown, but our findings show that by tuning simple parameters such as ionic strength or membrane composition, wetting transitions as well as complete engulfment of condensates can be induced.

The fact that changes in such simple system parameters allow for fine-tuning of condensate wetting and morphology suggests that cells might take advantage of local changes in cytosol and membrane composition to modulate organelle shape and interactions through wetting transitions. For example, wetting transitions have been recently reported to regulate the formation of the tight-junction belt via elongation of junctional condensates around the apical membrane

interface[60]. Furthermore, while epithelial junctions and focal adhesions[61] represent examples of complete wetting, processes such as the interaction of stress granules[13] or virus protein condensates[12,62] with the endoplasmic reticulum and the association of stress granules with lysosomes[63] appear to involve dewetting to partial wetting transitions. These examples underscore the diversity of cell processes involving wetting transitions as the result of the interaction between membrane-bound and membraneless organelles.

Under certain conditions, liquid condensates can turn into hollow condensates[6,50,64] that are believed to have a key role in the formation of multilayered membraneless organelles[50,64,65]. Here we show that membrane interaction with hollow condensates can provide additional compartmentalization for the system, with membrane or protein enclosed compartments. Membrane buds and necks can be formed as a result of the interplay between condensate-membrane adhesion, membrane elasticity, and capillary forces (Fig. 5, Supplementary Fig. 3), similar to deformations observed in storage vacuoles[57].

When excess membrane area is available, the droplet-membrane interface forms ruffles and fingers, generating irregularly curved structures (Fig. 6) characteristic of the glycinin-vesicle system, but not observed in ATPS-vesicle systems. The complex condensate-membrane morphologies observed for the protein condensates can be tuned by changing the membrane tension or by changing the material properties of the condensates. The diameters of the observed ruffles and fingers are around 1 μm and below (Fig. 6c, d, Supplementary Fig. 5), and thus are in the same order of magnitude of relevant mesoscale intracellular structures, such as the tubular network connecting the Golgi apparatus with the endoplasmic reticulum[66]. The membrane molding observed here suggests that interactions between membrane-bound organelles and condensates can trigger a complex mutual remodeling process, generating stable, prominently curved structures. In this context, it has been reported that during embryogenesis in plant cells, droplets of storage proteins, including glycinin, remodel the vacuolar membrane, generating finger-like structures that protrude into the cytoplasm[58] as shown in Supplementary Fig. 10d, e. In addition, glycinin condensation was proposed to occur in the endoplasmic reticulum in the range of ionic strengths used here[67] and could thus contribute to its morphology. This could also be a relevant mechanism for the morphogenesis and organization of membrane-bound organelles such as the Golgi apparatus[68,69]. Furthermore, these systems could provide templates for the design of multiphase compartments with complex connectivity for use in synthetic biology. Plant proteins like glycinin have the advantage of being widely abundant and economical, which makes them ideal materials for scaled-up processing in biotechnological applications[6]. Synthetic protocells can be sequentially assembled using microfluidics and pico-injection[70] to form droplet-stabilized GUVs which can then be loaded with different biomolecular components[71]. Injecting biomolecular condensates into GUVs can provide a promising method to create protocells that can undergo budding and form aqueous sub-compartments in a controlled and reproducible manner, allowing for the encapsulation and control of chemical or enzymatic reactions[72,73]. By changing the interfacial tension of the condensates, the size of these compartments can be varied over several orders of magnitude, from tens of nanometers to tens of micrometers. This system could also be exploited for the development of high-complexity synthetic biology systems, like the assembly of bacteriogenic protocells mediated by condensates[74]. Finally, condensate-membrane interactions triggering membrane ruffling and the formation of finger-like structures as shown here demonstrate that they do not only play a role in concentration-regulated phase separation and nucleation[9], but can also dramatically mold and shape membranes. Upon encounter with membrane-bound organelles, liquid condensates act as sculptors of intricate membrane structures, generating local curvature without the involvement of active processes.

## Methods

### Materials

The phospholipids 1,2-dioleoyl-sn-glycero-3-phosphocholine (DOPC), Soy L-α-phosphatidylcholine (soy-PC), 1,2-dipalmitoyl-sn-glycero-3-phosphocholine (DPPC), 1,2-dioleoyl-sn-glycero-3-phospho-L-serine (DOPS), and 1,2-dioleoyl-3-trimethylammonium-propane (DOTAP), cholesterol (Chol), 1,2-dipalmitoyl-sn-glycero-3-phosphatidylethanolamine-N-(lissamine rhodamine B sulfonyl) (DPPE-Rh) as solutions in chloroform and Neu5Acα2-3(Galβ1-3GalNAcβ1-4)Galβ1-4Glcβ1Cer (GM1) as a powder were purchased from Avanti Polar Lipids (IL, USA). The soluble dye, 2-(3-diethylamino-6-diethylazaniumylidene-xanthen-9-yl)−5-sulfo-benzenesulfonate (Sulforhodamine B) was obtained from Thermofisher (MA, USA). The fluorescent lipid dye, ATTO 647N-DOPE was purchased from ATTO-TEC GmbH (Siegen, Germany). Polyvinyl alcohol (PVA, with MW 145000) was purchased from Merck (Darmstadt, Germany). Chloroform obtained from Merck (Darmstadt, Germany) was of HPLC grade (99.8 %). Lipid stocks were mixed as chloroform solutions at 4 mM, containing either 0.5 mol% ATTO 647N-DOPE for the glycinin condensate experiments or 0.1 mol% DPPE-Rh for the ATPS experiments. Stocks were stored until use at −20 °C. For neutral lipid compositions in the measurements with glycinin, DOPC was employed, while charged membranes were prepared from DOPC:DOPS and DOPC:DOTAP mixtures. The membrane composition for the experiments with PEG-dextran ATPS was DOPC:GM1:DPPE-Rh, 95.9:4:0.1 mol%. Polydiallyldimethylammonium chloride (PDDA, 200-350 kDa, 20 wt% solution in H2O), adenosine triphosphate (ATP), fluorescein isothiocyanate isomer (FITC), sucrose, glucose, dimethyl sulfoxide (DMSO), Tris HCl buffer, potassium chloride (KCl), magnesium chloride (MgCl₂), sodium hydroxide (NaOH) and sodium chloride (NaCl) were obtained from Sigma-Aldrich (Missouri, USA). Gm1PEG (average molecular weight 8000 g/mol) and dextran from Leuconostoc mesenteroides (molecular weight between 400 kDa and 500 kDa) were purchased from Sigma-Aldrich. The oligopeptides, poly-L-lysine hydrochloride (degree of polymerization, $n = 10$; K10) and poly-L-aspartic acid sodium salt (degree of polymerization, $n = 10$; D10) were purchased from Alamanda Polymers (AL, USA) and used without further purification (purity≥95%). A N-terminal TAMRA-labeled version of K10 was purchased from Biomatik (Ontario, Canada). All solutions were prepared using ultrapure water from SG water purification system (Ultrapure Integra UV plus, SG Wasseraufbereitung) with a resistivity of 18.2 MΩ cm.

### Protein purification

Preparation of glycinin was achieved as described in Chen et al.[6]. Briefly, the defatted soy flour was dispersed 15-fold in water by weight and adjusted to pH 7.5 with 2 M NaOH. This slurry was then centrifuged at 9000 × g for 30 min at 4 °C. Dry sodium bisulfite (SBS) was added to the supernatant (0.98 g SBS/L), the pH of the solution was adjusted to 6.4 with 2 M HCl, and the obtained turbid dispersion was kept at 4 °C overnight. After that, the dispersion was centrifuged at 6500×g for 30 min at 4 °C. The glycinin-rich precipitate was dispersed 5-fold in water and the pH was adjusted to 7. The glycinin solution was then dialyzed against Millipore water for two days at 4 °C and then freeze-dried to acquire the final product with a purity of 97.5%[6].

### Protein labeling

A 20 mg/mL soy glycinin solution was prepared in 0.1 M carbonate buffer (pH 9). FITC was dissolved in DMSO at 4 mg/mL. The FITC solution was slowly added into the protein solution with gentle stirring to a final concentration of 0.2 mg/mL. The sample was incubated in the dark while stirring at 23 °C for three hours. The excess dye was removed using a PD-10 Sephadex G-25 desalting column (GE Healthcare, IL, USA) and the buffer was exchanged with ultrapure water. The pH of the labeled protein solution was adjusted to 7.4 by adding 0.1 M NaOH. For fluorescence microscopy experiments, an aliquot of this

solution was added to the working glycinin solution to a final concentration of 4%.

## Formation of glycinin condensates

A glycinin solution at 20 mg/mL was freshly prepared in ultrapure water and the pH was adjusted to 7. The solution was filtered to remove any insoluble materials using a 0.45 μm filter and was kept at 4 °C before use. To form the condensates, 50 μL of glycinin solution was mixed with the same volume of a NaCl solution of twice the desired final concentration, to obtain a 10 mg/mL solution of glycinin condensates. At this protein concentration, condensate size is optimal for microscopy and distances between condensates are big enough to prevent coalescence during the experiment.

## Hollow condensates

Salinity shifts in a coacervate suspension towards the phase-coexistence boundary, induced either by decreasing or increasing the salt concentration, result in hollow condensate formation by triggering the formation of protein-poor phases within already formed condensates, as previously reported[6]. Here, to generate hollow condensates, first 2 mL of soy glycinin solution containing 10% (w/w) FITC-labeled protein was mixed with an equal volume of 200 mM NaCl solution to induce condensate formation at a final concentration of 100 mM NaCl. This condensate suspension (100 μL) was then mixed with an equal volume of pure water to induce hollow condensate formation at a final salinity of 50 mM NaCl. To produce hollow condensates at 150 mM, the condensate suspension was mixed with an equal volume of 200 mM NaCl solution and then diluted with a 150 mM NaCl solution. These steps triggered phase separation within the preformed condensates, creating a protein-poor phase within them[6].

## PDDA/ATP droplet formation

Phase separated droplets were formed by gently mixing aliquots of stock solutions of Tris HCl (pH= 7.4), MgCl$_2$, glucose, PDDA and ATP (in this order) to a final volume of 20 μL. For labeling, a 0.5 mol% solution of the water-soluble dye, Sulforhodamine B was added. The final concentration of each component were as follows: 20 mM Tris HCl, 5 mM MgCl$_2$, 170 mM glucose, 14.8 mM ATP, and 4.9 mM PDDA. The final osmolality of the mixture was ≈ 200 mOsm/kg.

## Oligopeptides K10/D10 droplet formation

Phase separation was triggered by gently mixing aliquots of stock solutions of KCl, MgCl$_2$, glucose, D$_{10}$ and K$_{10}$ (in this order) to a final volume of 20 μL. For labeling, a 0.1 mol% solution of TAMRA-K$_{10}$ in water was added. The final concentration of each component were as follows: 15 mM KCl, 0.5 mM MgCl$_2$, 170 mM glucose, 2 mM D$_{10}$, and 2 mM K$_{10}$. The final osmolality of the mixture was ≈ 200 mOsm/kg.

## Lipid membranes

GUVs were grown using the electroformation method[75]. Briefly, 3–4 μL of a lipid stock was spread on two conductive indium tin oxide glasses and kept under vacuum for 1 h. The two glass electrodes were separated by a 2-mm-thick Teflon frame, forming the electroformation chamber. The lipid films were hydrated with 2 mL of a sucrose solution, matching the osmolality of the NaCl solution in which the condensates were formed. The osmolality was adjusted using a freezing point osmometer (Osmomat 3000, Gonotec). An electric AC field (1 V, 10 Hz, sinusoidal wave) was applied for one hour at room temperature. Once formed, vesicles were diluted 1:1 in a glucose solution of the same osmolality and the suspension of GUVs was stored at room temperature until use. The vesicles were prepared freshly before each experiment. For the experiments with PEG-dextran ATPS, the dried lipid films were hydrated in solutions of PEG-rich phase, as explained further below.

For the experiments in Supplementary Fig. 7, the GUVs were prepared with the PVA gel-assisted swelling method[76], which allows vesicle swelling in high salinity conditions. Briefly, two coverslips were cleaned with water and ethanol and dried under nitrogen. PVA solution was prepared by mixing PVA flakes with deionized water to a final concentration of 40 mg/mL and heating at 90 °C while stirring for 3 h. A small aliquot (20-50 μL) of the PVA solution was spread on the glass slides and dried for 1 h at 60 °C. A 3-4 μL layer of lipid stock solution was deposited on the PVA-coated glass and kept for 1 h under vacuum at room temperature. The chamber was assembled with a 2 mm-thick Teflon spacer and filled with 1 mL of 150 mM NaCl solution. After 30 min, the vesicles were carefully harvested in order to prevent PVA detachment from the cover glass.

## Condensate-membrane suspensions

Coverslips for confocal microscopy (26 × 56 mm, Waldemar Knittel Glasbearbeitungs GmbH, Germany) were washed with ethanol and water, then passivated with a 10 mg/mL BSA solution. A 1:10 dilution of the vesicle suspension was made in a diluent solution of the desired final NaCl concentration. The glycinin condensate suspension of the same NaCl concentration was diluted 1:4 with the NaCl diluent solution and added to the vesicle suspension at 15% v/v. After gently mixing the vesicle-condensate suspension, an aliquot was placed on a coverslip for imaging.

For experiments involving hollow condensates interacting with vesicles, 20 μL of the hollow condensate suspension was mixed with an equal volume of the 10-times diluted soy-PC vesicle solution of the same salt concentration.

For the vesicle deflation and inflation experiments, vesicles were electroformed in 200 mM, 300 mM, or 450 mM sucrose and then diluted in 150 mM NaCl. Condensates were prepared at 150 mM NaCl and mixed with the vesicle dilution as previously described. The vesicles were visually inspected for fluctuations and floppiness in the absence of condensates or interface ruffling when in contact with glycinin droplets. At least 60 vesicles were inspected in each condition and the experiment was performed three times.

For the interaction of membranes with PDDA/ATP or K$_{10}$/D$_{10}$ condensates, the vesicle suspension was diluted 1:10 in the final buffer of the corresponding droplet suspension. An aliquot of this diluted vesicle solution was then mixed with the droplet suspension in an 8:1 volume ratio directly on the cover glass and sealed for immediate observation under the microscope.

## PEG-dextran ATPS and membranes

Solution preparation and mixing with GUVs followed the procedure previously reported by our lab[34]. Briefly, a polymer solution composed of 6.87 wt % PEG and 2.86 wt % dextran (1.25 wt % of the total dextran was labeled with fluorescein isothiocyanate) was prepared and left for 2.5 days to completely phase separate and equilibrate. The vesicles were prepared in the PEG-rich phase and a small aliquot of the dextran-rich phase was mixed with the vesicle solution. The mixing chamber was gently shaken to break the dextran phase into small droplets. Imaging was done after 2 hours or on the next day.

## Confocal microscopy and FRAP

Confocal SP5 or SP8 microscopes equipped with 63×, 1.2 NA water immersion objectives (Leica, Mannheim, Germany) were used for imaging. FITC and ATTO 647N-DOPE were excited using the 488 nm and 633 nm laser lines, respectively.

FRAP measurements were performed on the SP8 setup equipped with a FRAP booster. A circular region of interest (ROI) with a diameter of 2 μm on the membrane was bleached with 3 iterative pulses of total time ~3 s. Fluorescence intensities from ROIs corresponding to photobleaching were analyzed using ImageJ. Curves were fitted using the

following formula: $y = (I_0 + I_{max}(x/\tau_{1/2}))/(1 + x/\tau_{1/2})$, where $I_{max}$ is the maximal intensity and $\tau_{1/2}$ is the halftime of recovery.

## STED microscopy

An Abberior STED setup (Abberior Instruments GmbH) based on an inverted Olympus IX83 microscope (Olympus Inc., Japan) equipped with a 60×,1.2 NA water immersion objective was used to obtain the super-resolved images. The sample was excited at 640 nm and a 775 nm pulsed beam was used as the depletion laser. Alignment was achieved as described previously for the setup[43]. Briefly, 150 nm gold beads (Sigma-Aldrich, USA) were observed in reflection mode to overlap the center of the excitation focus with the center of the depletion focus. Corrections for mismatches between the scattering and fluorescence modes were performed using 100 nm TetraSpeck™ beads (Invitrogen, USA). To measure the resolving power of the setup, crimson beads of 26 nm diameter (FluoSpheres™, Molecular Probe) were used. A measured resolution of ~35 nm was achieved using 80% STED laser power (total laser power of 1.25 W), improving by 10-fold the lateral resolution of the corresponding excitation laser[43]. For our experiments, 3D STED was more suitable than 2D STED, since we could eliminate the interference of out-of-focus signal coming from the curved regions of the membrane (see Supplementary Fig. 5). For the images shown in Fig. 6c and Supplementary Fig. 5, the pixel size is 50 nm and the pixel dwell time is 10 μs.

## Micropipette aspiration

Micropipettes were formed by pulling glass capillaries (World Precision Instruments Inc.) with a pipette puller (Sutter Instruments, Novato, CA). Pipette tips were cut using a microforge (Narishige, Tokyo, Japan) to obtain smooth tips with inner diameter between 6-10 μm. The pipette tips were coated with a 2 mg/mL solution of casein (Sigma) to prevent adhesion. Latex microspheres of 6 μm diameter (Polysciences Inc., PA, USA) were used to determine the zero-pressure level. The aspiration pressure was controlled through adjustments in the height of a connected water reservoir mounted on a linear translational stage (M-531.PD; Physik Instrumente, Germany). Images were analyzed using ImageJ software.

## Wetting geometries and contact angles

All wetting geometries involve three aqueous solutions: the interior solution, $i$ within the vesicle; the exterior buffer, $e$; and the glycinin condensate $c$. These aqueous solutions are separated by three surface segments (Fig. 2b). The glycinin condensate, $c$ is separated from the exterior buffer, $e$ by the $ce$ interface. When the interface forms a contact line with the membrane, this line partitions the membrane into two membrane segments. The membrane segment $ic$ forms the contact area with the condensate whereas the other membrane segment, $ie$ is exposed to the exterior buffer. At the contact line, the three surface segments form three apparent contact angles, $\theta_c$, $\theta_i$, and $\theta_e$ that add up to 360° (Fig. 2b). These three contact angles are related to the three surface tensions via the tension triangle in Fig. 2c[52] and to the affinity contrast, $W$ as defined in Eq. (9) below.

## Contact angle determination

In order to adequately measure the contact angles between the different interfaces from microscopy projections, it is necessary for the rotational axis of symmetry of the vesicle-droplet system to lie in the image plane of the projected image (see Supplementary Fig. 11). Otherwise, an incorrect projection will lead to misleading values of the system geometry and contact angles. Once the 3D image of the vesicle and the droplet is acquired and reoriented to obtain correct projection, we consider the three spherical caps of the vesicle, the droplet, and the vesicle-droplet interface and fit circles to their contours to extract the different radii, $R_i$, and center positions, $C_i$, as defined in Supplementary Fig. 11b. In this manner, all contact angles can be

defined by the following expressions:[37]

$$\sin \theta_i = \frac{R_4}{R_1 R_3}\left(\sqrt{R_3^2 - R_4^2} + \sqrt{R_1^2 - R_4^2}\right) \quad (1)$$

$$\sin \theta_c = \frac{R_4}{R_3 R_2}\left(\sqrt{R_3^2 - R_4^2} - \sqrt{R_2^2 - R_4^2}\right) \quad (2)$$

with $R_3 \geq R_2$

$$\sin \theta_e = \frac{R_4}{R_1 R_2}\left(\sqrt{R_1^2 - R_4^2} + \sqrt{R_2^2 - R_4^2}\right) \quad (3)$$

For the case where the center position $C_3$ is located above $C_1$ (see Supplementary Fig. 11b) and the membrane is curved towards the droplet, these relationships become:[37]

$$\sin \theta_i = \frac{R_4}{R_1 R_3}\left(\sqrt{R_3^2 - R_4^2} - \sqrt{R_1^2 - R_4^2}\right) \quad (4)$$

With $R_3 \geq R_1$

$$\sin \theta_c = \frac{R_4}{R_3 R_2}\left(\sqrt{R_3^2 - R_4^2} + \sqrt{R_2^2 - R_4^2}\right) \quad (5)$$

and

$$\sin \theta_e = \frac{R_4}{R_1 R_2}\left(\sqrt{R_1^2 - R_4^2} + \sqrt{R_2^2 - R_4^2}\right) \quad (6)$$

where $R_1$, $R_2$, and $R_3$ are the radii of the circles fitting the vesicle, the droplet, and the contact line, respectively. $R_4$ is the apparent contact line radius, which can be obtained as follows:

$$R_4 = \frac{2}{d_{12}}\sqrt{s(s - d_{12})(s - R_2)(s - R_1)} \quad (7)$$

where $s = \frac{R_1 + R_2 + d_{12}}{2}$. In this manner, the only input parameters required to be measured from the projection images are the radii of the fitted circles, $R_1$, $R_2$, and $R_3$, and the distance between the centers of the vesicle and the droplet circles, $d_{12}$.

Note that the above analysis is only valid for spherical cap morphologies as shown in Figs. 2–3 and cannot be applied to ruffled membrane-droplet interfaces (Figs. 6–8), which do not fulfill the spherical cap condition.

## From contact angles to fluid-elastic parameters

The contact angles in Figs. 2–4, are related to the different surface tensions that pull the three surface segments along the contact line (see Fig. 2b,c). One of these surface tensions is provided by the interfacial tension, $\Sigma_{ce}$ of the condensate-buffer interface. The latter tension is balanced by the difference, $\Sigma_{ie}^m - \Sigma_{ic}^m$ between the membrane tensions, $\Sigma_{ie}^m$ and $\Sigma_{ic}^m$ of the two membrane segments. This balance implies that the three surface tensions form the sides of a triangle (Fig. 2c). As previously shown[37], the mechanical tensions of the two membrane segments are given by

$$\Sigma_{ie}^m = \Sigma + W_{ie} \quad \text{and} \quad \Sigma_{ic}^m = \Sigma + W_{ic} \quad (8)$$

where $\Sigma$ denotes the lateral stress within the membrane, which is conjugate to the total surface area, $A$ of the vesicle membrane, whereas $W_{ie}$ and $W_{ic}$ represent the adhesion free energies per unit area of the external buffer, $e$, and of the condensate, $c$, relative to the interior solution, $i$[37]. The decomposition in Eq. (8) follows from the shape functional for vesicle-droplet systems, which contains both the

Lagrange multiplier term, $\Sigma A$ and the adhesion energy, which depends on the adhesion parameters, $W_{ie}$ and $W_{ic}$. The adhesion parameter, $W_{ic}$ is negative if the membrane prefers the condensate over the interior solution and positive in the opposite case. For simplicity, possible contributions from the spontaneous curvatures of the membrane segments[37] have been ignored in Eq. (8). The affinity contrast between the condensate and the external buffer is then given by

$$W = \Sigma_{ic}^m - \Sigma_{ie}^m = W_{ic} - W_{ie} \text{ with } -\Sigma_{ce} \leq W \leq +\Sigma_{ce} \quad (9)$$

where the inequalities follow from the tension triangle in Fig. 2c and the general property that each side of a triangle must be smaller or equal to the sum of the two other sides. Note that the geometry-dependent lateral stress, $\Sigma$ drops out from the affinity contrast, $W$, which is negative if the membrane prefers the condensate phase, $c$ over the external buffer, $e$, and positive otherwise. The limiting value $W = -\Sigma_{ce}$ corresponds to complete wetting by the condensate phase whereas the limiting case $W = +\Sigma_{ce}$ describes dewetting from the condensate phase, which is equivalent to complete wetting by the external buffer.

The tension triangle in Fig. 2c also implies the relationships[37]

$$\frac{\Sigma_{ie}^m}{\Sigma_{ce}} = \frac{\Sigma + W_{ie}}{\Sigma_{ce}} = \frac{\sin\theta_c}{\sin\theta_i} \text{ and } \frac{\Sigma_{ic}^m}{\Sigma_{ce}} = \frac{\Sigma + W_{ic}}{\Sigma_{ce}} = \frac{\sin\theta_e}{\sin\theta_i} \quad (10)$$

between the surface tensions and the contact angles, as follows from the law of sines for the tension triangle in Fig. 2c. When we take the difference of the two equations in Eq. (10), the affinity contrast, $W$ in Eq. (9) becomes equal to

$$W = \Phi\Sigma_{ce} \text{ with } \Phi \equiv \frac{\sin\theta_e - \sin\theta_c}{\sin\theta_i} \quad (11)$$

Thus, the rescaled affinity contrast, $W/\Sigma_{ce}$, which is a mechanical quantity related to the adhesion free energies of the membrane segments, is equal to the factor, $\Phi$, which is a purely geometric quantity that can be obtained from the three contact angles, as determined by analysing the optical images (Fig. 2a,e). The inequalities in Eq. (9) imply the inequalities $-1 \leq \Phi \leq 1$ for the geometric factor, $\Phi$, where the interpretation for the limiting values of $\Phi$ follows from the limiting cases for the affinity contrast, $W$ in Eq. (9). The smallest possible value of $\Phi = -1$ corresponds to complete wetting of the membrane by the condensate phase, while the largest possible value of $\Phi = +1$ corresponds to dewetting of the membrane by this phase. The dimensionless factor, $\Phi$ is negative if the membrane prefers the condensate over the exterior buffer and positive otherwise. It then follows from the data in Fig. 2d that the membrane prefers the condensate for molar salt concentrations $X > X_0 \simeq 93$ mM and the exterior buffer for $X < X_0$. For $X = X_0$, the geometric factor, $\Phi$ vanishes, corresponding to equal contact angles, $\theta_c = \theta_e$ and no affinity contrast between the condensate and the exterior buffer, $W = 0$. Therefore, by measuring the geometric factor, $\Phi$ as a function of the salinity or another control parameter, we obtain the affinity contrast, $W = \Sigma_{ic}^m - \Sigma_{ie}^m = \Phi\Sigma_{ce}$ in units of the interfacial tension, $\Sigma_{ce}$ as a function of this control parameter.

Note that the geometric factor is scale-invariant and does not depend on the sizes of a given vesicle-condensate couple, as exemplified in Supplementary Fig. 12.

## Criterion for ruffling of the condensate-membrane interface

Ruffling and formation of finger-like structures at the *ic* interface implies an increase of the bending energy, assuming negligible spontaneous curvature. Ruffling will be observed when this bending energy is overcompensated by the gain in adhesion energy for transferring membrane area from the condensate-free *ie* segment to the *ic*

interface in contact with the condensate. When we transfer the membrane area, $\triangle A$, the gain in adhesion energy is $E_{ad} = W\triangle A$. Introducing Eq. (11) yields $E_{ad} = \Phi\Sigma_{ce}\triangle A$, which is negative for negative $\Phi$; note that ruffling is observed only at for [NaCl] $\geq 100$ mM, where $\Phi < 0$ (see Fig. 1). The bending energy, $E_{be}$ is proportional to $8\pi\kappa$ and also proportional to the number, $N_{pro}$ of protrusions formed by the membrane area, $\triangle A$. The total energy change associated with the transferred membrane area is then given by $\Delta E = E_{ad} + E_{be} = \Phi\Sigma_{ce}\triangle A + c8\pi\kappa N_{pro}$. Here, $\Phi < 0$ and $c$ is a dimensionless coefficient of the order of 1. This energy change, $\Delta E$ is negative for $|\Phi|\Sigma_{ce}\triangle A > c8\pi\kappa N_{pro}$, where $|\Phi|$ is the absolute value of $\Phi$. Ignoring the dimensionless coefficient, $c$, we obtain the simple criterion

$$|\Phi|\Sigma_{ce}\triangle A > 8\pi\kappa N_{pro} \quad (12)$$

for the area transfer and ruffling to be energetically favorable. To examine this relationship, we consider the vesicle-condensate couple shown in Fig. 8b. For the interfacial tension and the bending rigidity, we take $\Sigma_{ce} \simeq 0.5$ mN/m, as measured here, and $\kappa \simeq 10^{-19}$ J. The area change stored in ruffles that can be pulled out with a micropipette is of the order of 50 $\mu m^2$. For the salinity conditions where ruffling is observed ([NaCl] $\geq 100$ mM), $|\Phi|$ ranges between 0.2 and 1 (see Fig. 1). If we take for the number of protrusions, $N_{pro} = 100$, which is a generous overestimate, we still obtain that the criterion, Eq. (12) is satisfied whereby the adhesion energy gain is still orders of magnitude higher than the bending energy penalty for ruffling.

Judging from the images in Figs. 6–8, the fingers and undulations do not have a characteristic length scale. Their dimensions are defined by the membrane area, $\triangle A$ available to form the protrusions, as well as the volume constraints imposed by the sizes of the condensate and vesicle.

## Glycinin condensate material properties

Measuring the material properties of biomolecular condensates is a challenging task that typically relies on fluorescence recovery after photobleaching (FRAP) measurements and on quantifying the coalescence kinetics of two condensate droplets[2,77,78]. These techniques can provide information about the viscosity[77,78] and can yield the inverse capillary number, or the ratio of surface tension to viscosity[2], respectively. Recently, the micropipette aspiration method has been applied to quantify both the viscosity and the surface tension of condensate droplets, provided that they behave as Newtonian fluids at the experimental time scale (>1 s)[47].

In a previous study, glycinin condensates were shown to exhibit only negligible fluorescence recovery in FRAP experiments, suggesting a highly viscous environment[6]. By using the condensate coalescence assay, the inverse capillary number of glycinin droplets was measured to be $\eta/\Sigma_{ce} \approx 9.69$ s/$\mu$m[6]. A very rough estimate of the surface tension based on the molecular size of glycinin yielded a value of $\Sigma_{ce} \approx 0.16$ mN/m, implying a viscosity of $\eta \approx 1.6$ kPa.s[6].

Here, we attempted to measure directly the viscosity and surface tension by means of the micropipette aspiration method. However, as can be observed in Supplementary Fig. 13, glycinin condensates do not flow inside the pipette. The aspiration can only proceed until a certain pressure value, beyond which the pipette gets clogged and the condensate cannot be further aspirated nor released. The maximum suction pressure achieved by our setup is 2500 Nm$^{-2}$. This outcome does not depend on the condensate/pipette diameter ratio. In addition, aging effects were excluded, since they occur at time scales on the order of days[6,79], much longer than those required for our experiments which occur on the order of minutes to hours. Pipette clogging prevented us from measuring the viscosity, but we could estimate the tension by means of the Laplace equation, $\Sigma_{ce} = \triangle PR_{cap}/2(1 - \frac{R_{cap}}{R_{drop}})$, where $\triangle P$ is the applied pressure and $R_{cap}$ and $R_{drop}$ are the radius of the spherical cap formed by the part of the droplet inside the pipette

and the radius of the droplet outside the pipette, respectively (see sketch in Supplementary Fig. 13b). The tension value obtained in this way for glycinin droplets in the presence of 100 mM NaCl is $\Sigma_{ce} = (0.5 \pm 0.3)$ mN/m ($n = 5$), which is on the same order of magnitude as the value previously estimated from the coalescence assay. The condensate viscosity we obtain is $\eta \approx 4.8$ kPa.s. The surface tension of glycinin condensates is similar to that of LAF-1[80] and PolyR[81] condensates and the viscosity is close to that of the nucleolus[82], as summarized in Fig. 1 of the work of Wang et al[47]. This particular combination of high surface tension and high viscosity might be related to the structural characteristics of these condensates. While more coacervation-prone proteins feature mostly one type of polypeptide[83,84], glycinin is a hexamer[85,86] implying that it is characterized by a more bulky and complex structure. This could explain the reduced diffusion and high viscosity of the condensates.

## Statistics and reproducibility
All experiments were repeated at least three times, obtaining reproducible results. Images shown are representative of the sample within the particular conditions described. Mean values, standard deviations, and n numbers are indicated for each plot.

## Reporting summary
Further information on research design is available in the Nature Portfolio Reporting Summary linked to this article.

## Data availability
The source data underlying Figs. 2d, 3b, 4b,d, 7d, 8a as well as Supplementary Figs. 1d, 2b, and 12 are provided in a separate Excel file labeled 'Source Data'. Source data are provided with this paper.

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

## Acknowledgements

A.M. acknowledges support from Alexander von Humboldt Foundation. N.C. acknowledges funding from the National Natural Science Foundation of China (No. 32101972). Z.Z. acknowledges support from Grant SarsRapid by the Bank of Thuringia and CRC 1278 Polytarget. We acknowledge Y. Li for sharing images of vesicles in the presence of PEG-dextran ATPS. Some parts of this study were inspired by work performed in the MaxSynBio consortium, which was jointly funded by the Max Planck Society and the German Federal Ministry of Education and Research (BMBF). In particular, R.L. and R.D. are thankful for discussions and insightful comments on the manuscript by A.A. Hyman and T.M. Franzmann. We acknowledge N. Tam for proofreading the text.

## Author contributions

A.M. performed most of the experiments. N.C. purified the protein, performed the experiments on hollow condensates, and pilot experiments to find the optimum conditions for condensate-membrane interaction. Z.Z. aided with STED imaging and provided ATPS contact angle measurements. A.M. analyzed the data. R.L developed the theoretical framework. R.D. supervised the project. A.M., R.L., and R.D. wrote the paper, with input from the rest of the authors.

## Funding

## Competing interests

All authors declare no competing interests.
