## [Peer Review File · Nature Communications]

REVIEWER COMMENTS

Reviewer #1 (Remarks to the Author):

The study discusses how interactions between lamellar membranes of GUVs and protein condensates can give rise to complex morphological changes in the membrane via broad regimes of wetting-dewetting processes effected by salt concentrations and membrane lipid charges. Although the experimental evidence is solid, it does not show the physiological relevance in the context of cellular membranes and proteins.

Major concerns:

- 1) The data presented are in vitro without showing the implications in the cellular context. Experiments to use the vesicles isolated from the cells or directly visualization in vivo should be performed to validate the observation. It would also be highly beneficial if the authors can show such membrane ruffling events leading to the vesicle formation within the cells and whether such changes are affected by altering the surface tension by changing osmolarity and chemical methods.
- 2) The timescale of membrane ruffling due to wetting is in the range of tens of minutes to hours while within cells, vesicle formation takes place in milliseconds to seconds to minutes timescale.
- 3) The membranes (GUVs) used to examine the interactions with condensates were constituted of a homogeneous mixture of uncharged or charged lipids i.e. DOPS, DOTAP, or Phosphatidylcholine. The authors should address how the compositional complexity of membranes might impact membrane-condensate interactions. For this, membranes with mixtures of different charged and uncharged lipids added in various stoichiometries should be used to decipher the effects.
- 4) Can the two-phase transitions from dewetting to partial wetting to complete wetting be shown for other proteins or heteropolymers with different charge distributions or with a mixture of proteins. Can it be shown for example with viral capsid proteins or RNA-binding proteins? Also, it would be relevant to plot the changes in the geometric factor for the PEG-dextran system for comparison in figure 2.
- 5) Would the force distribution follow similar patterns if the size of the GUVs is smaller than the condensates?
- 6) it would be nice to have theoretical model to strengthen their conclusions. But I understand this will be a tremendous effort and thus it is authors' choice.

Minor Concerns:

- 1) The manuscript has some typos needed to be corrected.
- 2) In figure 5, what determines whether the membrane condensate mixture would be involved in bridging or would undergo an active engulfment procedure?
- 3) What are the parameters and thresholds used to consider the separate phases of dewetting, partial and complete wetting?
- 4) All images in a figure panel should have the same scale for a better understanding of the readers. This needs to be corrected for all figures. The addition of scale bar in all the images is needed to make the figures consistent.
- 5) It would be good if the schematics of the experimental workflow are added to the figures before representing the results.
- 6) In figure 3b, the number of experiments is not mentioned for SD calculations.

Reviewer #2 (Remarks to the Author):

This manuscript uses a model in vitro system comprised of biomolecular condensates

formed by the protein glycinin, together with membrane vesicles, to examine the biophysical basis of condensate-membrane wetting behavior. The authors show that there is a salt concentration dependent changes in this wetting behavior, which can be tuned from completely non-wetting to completely wetting. They report and examine a number of interesting feature, including a surprising slow relaxation to equilibrium contact angles, as well as a differential mobility of membranes on the different interfaces (the condensate wetting appears to significantly slow membrane mobility). Probably most strikingly, the authors describe a fascinating behavior in which the membrane ruffles, and use micropipette aspiration experiments to show how this appears to be related to excess membrane availability and associated membrane tension.

Overall, I find this to be a very solid manuscript, that examines a timely and interesting problem, and thus I think it is quite suitable for publication in Nature Communications. I have only relatively minor suggestions that can be incorporated in the revised manuscript.

Minor comments:

1. The authors refer to the data as describing “three different wetting regimes separated by two wetting transitions”. Is it really appropriate to refer to these as different wetting regimes, when ultimately this is behavior that can be described by a single continuous variable (ϕ)?
2. The authors describe the membrane kink as reflecting $\sqrt{\kappa/\sigma_{ce}}$, but don’t use the term “Elastocapillary length”. Is there a reason for this? I would suggest calling it that, is it is a pretty well-known name for that quantity.
3. In the methods, it says “..and the viscosity, close to that of the nucleus”; this should be “nucleolus”.
4. I found the description of the physics underlying the ruffling behavior somewhat unclear/unsatisfying. Perhaps the authors could say a bit more, in particular about what sets the length scale of ruffling. This is presumably also related to the elastocapillary length, although the curvatures here appear to be smaller than the high curvature of the “kink” regions at the contact lines.

ANSWERS TO REVIEWERS' COMMENTS

(Reviewers' comments are in black and our answers in blue)

Reviewer #1:

The study discusses how interactions between lamellar membranes of GUVs and protein condensates can give rise to complex morphological changes in the membrane via broad regimes of wetting-dewetting processes effected by salt concentrations and membrane lipid charges. Although the experimental evidence is solid, it does not show the physiological relevance in the context of cellular membranes and proteins.

We thank the Reviewer for the comments and suggestions, and for considering our work solid. Below we address the concerns point by point. The Reviewer's comments prompted us to include a more detailed discussion on the biological relevance of our findings (see below), which we have now done expanding the discussion section. Additionally, we have conducted new experiments (both with more complex membrane compositions and with different condensate systems) that underscore how widely applicable and universal our findings are. We also strengthened and extended the theoretical description and the model proposed for condensates-membrane interactions. New data and figures were included in the supporting material (see Figs. 9, S1b-d, S4b, S5b, S8, S9 and S11, and also included here below). We believe the clarity of the manuscript has improved, and the biological implications of our findings are now highlighted.

Motivated by the Reviewer's comments, we now review studies in cells that highlight the relevance of our findings *in-vivo*. We also assembled a new figure (Figure 9) including these findings and added the following text in the discussion section:

*"Glycinin is a major storage protein in the soybean and plays a key role in plant embryogenesis. In most cotyledons, storage proteins are accumulated in the vacuoles¹, and recently it has been reported that they not only provide nutrients for subsequent growth but also contribute to the tonoplast (vacuolar membrane) remodeling during plant development². Storage proteins form phase-separated droplets in the vacuole interior that interact with membranes presenting different wetting states, as is visible in electron microscopy images^{3, 4} (see Fig. 9a-e), and with fluorescence confocal microscopy² (Fig. 9f). In the final development stage, the protein droplets undergo complete engulfment by the vacuolar membrane, forming protein bodies³ (Fig. 9c). Such remodeling processes are relatively slow (days) as demonstrated for protein storage vacuoles in *Arabidopsis thaliana*², presumably related to the high viscosity of storage protein droplets (also shown here for glycinin). The precise mechanisms by which these processes occur in the plant cells remain unknown, but our findings show that by tuning simple parameters such as the ionic strength or the membrane composition, wetting transitions as well as complete engulfment can be induced.*

Figure 9. Wetting and remodeling of the tonoplast (vacuolar membrane) by storage protein condensates in plants. Electron micrographs of storage parenchyma cells of soybean cotyledons during development (a-e). Abbreviations: p=protein droplets, p_b=protein bodies, s=starch grains, v=vacuole, w=cell wall. **(a)** Protein droplets form a thin layer on the vacuolar membrane (see darker areas to which the yellow arrows point). **(b)** Partial wetting of the protein droplets on the vacuolar membrane. Note that the membrane acquires additional curvature in the contact regions with the condensate droplets. **(c)** Protein bodies are formed as a result of complete engulfment of the protein droplets by the tonoplast and subsequent pinching off. **(d)** “Protein pockets” are formed in the vacuoles visualized as finger-like structures protruding to the cytoplasm. This is proposed to be a step prior to protein body formation³. The inset shows the zoomed dotted region, and the protrusion is highlighted by an orange dotted line. **(e)** Another example of protrusion formation in the vacuolar membrane. **(f)** Confocal microscopy images of protein storage vacuoles in *Arabidopsis thaliana* embryo cells showing how the protein droplets (red) wet the vacuolar membrane (green). Scale bars in (a-e) are 1 μm and in (f) 10 μm. Images (a-e) were adapted from reference³ with permission from SNCSC and image (f) from reference².

Recently, wetting transitions have been reported to regulate the formation of the tight-junction belt via elongation of junctional condensates around the apical membrane interface⁵. This is yet another example that underscores the diversity of cell processes involving wetting transitions as the result of the interaction between membrane-bound and membranellar organelles.”

“...The membrane molding observed here suggests that interactions between membrane-bound organelles and condensates can trigger a complex mutual remodeling process generating prominently curved and stable structures. In this context, it has been reported that during embryogenesis in plant cells, droplets of storage proteins, including glycinin, remodel the vacuolar membrane generating protrusions that are finger-like structures inserting in the cytoplasm³ as shown in Fig. 9d,e.”

Major concerns:

1) The data presented are *in vitro* without showing the implications in the cellular context. Experiments to use the vesicles isolated from the cells or directly visualization *in vivo* should be performed to validate the observation. It would also be highly beneficial if the authors can show such membrane ruffling events leading to the vesicle formation within the cells and whether such changes are affected by altering the surface tension by changing osmolarity and chemical methods.

We agree with the Reviewer that having new *in-vivo* data (apart from the one we now summarize in Fig. 9) or experiments with natural membranes would reinforce the findings of our work. Following the recommendations, we performed experiments using giant plasma membrane vesicles (GPMVs) isolated from live cells. However, as shown in the images below, the interaction between the glycinin droplets and the membrane was precluded (dewetting), presumably because of (i) lipid scrambling and exposure of negative charges on the vesicle surface during extraction which drives the system to the dewetted state, (ii) high density of the glycocalyx in the outer layer, and (iii) probably the fact that our cells are mammalian cells while glycinin is a plant protein. Despite this, we have conducted additional *in-vitro* experiments varying the membrane and the condensate compositions (as more precisely specified in the responses below) included in new Fig. S1b-d, S4b, S5b, S8 and S9, which confirm the presence of the wetting transitions and membrane ruffling, underscoring the broad applicability of our observations.

Examples of GPMVs (red, labeled with Rhodamine-DOPE) incubated with glycinin droplets (green, labeled with FITC) in solution containing 10 mM HEPES, 150 mM NaCl, 2 mM CaCl₂, (pH=7.4, ~300mOsm). The GPMVs were extracted from HEK cells using two protocols following Ref.⁶, namely by using 2 mM N-ethyl maleimide (first image) or 25 mM paraformaldehyde/2mM dithiothreitol (second image).

2) The timescale of membrane ruffling due to wetting is in the range of tens of minutes to hours while within cells, vesicle formation takes place in milliseconds to seconds to minutes timescale.

We believe that the timescale of the process of interfacial ruffling is related to the particular material properties of the condensate. Glycinin is a hexamer of high molecular weight (360kDa) and the condensates are highly viscous as shown in our measurements with micropipettes ($\eta \approx 4.8$ kPa.s.) and as suggested by earlier coalescence dynamics studies⁷. This value is in the high-viscosity range for already explored molecular condensates as shown in Wang et al.⁸. For example, it is three orders of magnitude higher than that of condensates formed by proteins such as FUS (0.7 Pa.s.) or PGL-1 (1 Pa.s.)⁸, see for example Fig 1 in Wang et al.⁸. Correspondingly, the dynamics of deformation should be slowed down as well. The times scales mentioned by the Reviewer do not necessarily relate to condensate-mediated vesicle formation. Our time scales are corroborated by studies demonstrating that the remodeling of protein storage vacuoles in *Arabidopsis thaliana* is a very slow and asynchronous process that can take up to several days². The slow dynamics of this cellular process could be as well related to the high viscosity of storage protein droplets.

To address this comment, we modified the text on page 7 as follows:

“...Once in contact, the angles characterizing the condensate-membrane system required 20-30 min to reach equilibrium (Fig. 4b). This is most likely related to the high molecular weight of glycinin (360 kDa) that translates in the high viscosity of glycinin condensates ~ 4.8 kPa s (see Methods), which is roughly three orders of magnitude higher than that of condensates formed by other proteins like FUS (0.7 Pa.s.) or PGL-1 (1 Pa.s).”

The slow dynamics was also addressed in on page 14:

*“Such remodeling processes are relatively slow (days) as demonstrated for protein storage vacuoles in *Arabidopsis thaliana*², which might be related to the high viscosity of storage protein droplets (also shown here for glycinin).”*

3) The membranes (GUVs) used to examine the interactions with condensates were constituted of a homogeneous mixture of uncharged or charged lipids i.e. DOPS, DOTAP, or Phosphatidylcholine. The authors should address how the compositional complexity of membranes might impact membrane-condensate interactions. For this, membranes with mixtures of different charged and uncharged lipids added in various stoichiometries should be used to decipher the effects.

While several recent studies have already addressed the effect of membrane composition and phase state (and phase coexistence) on the interaction with condensates^{9, 10}, as well as the coupling between lipid domains and condensates^{11, 12}, we focused on a more poorly investigated topic, namely understanding the mechanism of membrane wetting and remodeling by condensates. Since this subject has not been explored, we initiated the studies using single phospholipid membranes, to first evaluate the effect of the milieu on wetting as in Figures 2 and S1. To evaluate the effect of membrane charge we employed binary mixtures at different stoichiometries of DOPC and a negatively charged phospholipid (DOPS) as well as the positively charged lipid (DOTAP). From our results (Figs. 3 and S2) it is clear that the membrane composition has a direct impact on the wetting state. However, it is important to note that this wetting state will depend on the particular protein condensate, since as mentioned previously, every condensate will have different material properties and, in addition, will exhibit different specific interactions with particular lipids. For glycinin, the inclusion of charges in the membrane promotes dewetting, however for other proteins the presence of charges could favor the wetting, as in the case of Dengue and Zika virus capsid proteins, that associate with DNA/RNA and wet negatively charged membranes¹³. This variability is also demonstrated by our new data with PDDA/ATP and K_{10}/D_{10} condensates, which we discuss in the response to the next question of the Reviewer.

Besides clarifying this aspect in the main text (see highlighted changes in the Introduction and Discussion sections), we performed additional experiments to expand the explored membrane compositions by using a ternary mixture containing cholesterol, DOPC:DPPC:cholesterol (64:15:21); this membrane composition has been previously used in ATPS systems¹⁴ which is our reference point. We have not only observed the two wetting transitions with this new membrane composition, but also the interfacial ruffling, which we characterized with confocal and super-resolution microscopy as we did for pure DOPC membranes. The new data is included as new panels Fig. S1c,d, S4 and S5b (see combined new panels below). When compared to simpler DOPC membranes, the observed differences in the wetting transitions are small, see comparison of the geometric factors in the figure below (Fig. S1d). These results indicate that introducing certain complexity and adding fast-flipping cholesterol has only a minor effect in tuning wetting compared to effects triggered by changing the membrane surface charge or environmental salinity, for this particular condensate-membrane system.

Combined new panels from **Figs. S1c,d, S4 and S5b** showing that **wetting transitions and interfacial ruffling** are observed for the ternary mixture **DOPC:DPPC:Chol (64:15:21)**. **(Fig. S1c)** Confocal microscopy images showing different wetting morphologies for the DOPC:DPPC:Chol GUVs (same membrane composition as in b) in contact with glycinin condensates at the indicated NaCl concentrations. Scale bars: 5 μm . **(Fig. S1d)** Geometric factor, Φ vs NaCl concentration for DOPC:DPPC:Chol GUVs in contact with glycinin (black, red and blue circles) and the data for pure DOPC GUVs shown in Fig. 2 (green diamonds). The green shadowed area corresponds to the SD for the DOPC data from Fig. 2d in the main text. Note that the data for the ternary mixture is shifted to slightly higher NaCl concentrations compared to that for pure DOPC. The red and blue circles correspond to the respective transitions from dewetting to partial wetting and from partial wetting to complete wetting for the DOPC:DPPC:Chol membrane. All data: mean \pm SD, $n=10$ per composition. **(Fig. S4b)** Interfacial ruffling is observed for the ternary mixture. Curving is visible on the membrane and the condensate. Scale bars: 5 μm . **(Fig. S5b)** Zoomed images of the membrane region delimited in the first micrograph. 3D STED allows for resolving the curved structures on the membrane. Scale bars: 1 μm . On the right: intensity profiles along the indicated dashed line in the STED image showing the increased resolution of STED microscopy and allowing us to estimate the thickness of the curved structures.

4) Can the two-phase transitions from dewetting to partial wetting to complete wetting be shown for other proteins or heteropolymers with different charge distributions or with a mixture of proteins. Can it be shown for example with viral capsid proteins or RNA-binding proteins? Also, it would be relevant to plot the changes in the geometric factor for the PEG-dextran system for comparison in figure 2. We thank the Reviewer for these suggestions. Our results and findings can be indeed extended to other phase-separating proteins or polymers. One very recent example *in vivo* is the wetting transition observed

for zona occludens proteins mediated by the apical protein PATJ, in the formation of tight junctions⁵. To provide results also *in vitro*, we have investigated the membrane interaction with droplets made of a mixture of a polyamine, (poly-diallyldimethylammonium chloride, PDDA) and ATP (we chose these systems rather than those suggested by the Reviewer, because conditions for preparing these condensates and partial knowledge of the phase diagram are available). These coacervates have been used previously as models for endocytosis¹⁵, synthetic cells¹⁶ and synthetic biology developments towards understanding the origin of life¹⁷. We observed that by increasing the negative charge of the membranes, the system undergoes two wetting transitions, from dewetting to partial wetting and complete wetting:

Figure S8. Wetting transitions for pure DOPC or DOPC:DOPS GUVs in contact with PDDA/ATP droplets. (a) Chemical structures of poly-diallyldimethylammonium chloride (PDDA, 200-350 kDa), and adenosine triphosphate (ATP). (b) Examples of confocal fluorescence and bright-field images of PDDA/ATP coacervates labeled with Sulforhodamine B. (c) Confocal microscopy images of PDDA/ATP droplets (green) in contact with DOPC vesicles (magenta) containing different molar fraction DOPS as indicated above the images. Two wetting transitions are observed for this system when changing the percentage of anionic lipids in the membrane, whereby complete wetting occurs at high fraction of the charged lipid, contrary to the behavior observed with glycinin (compare to Fig. 3a). Scale bars: 5 μm .

Note that while for glycinin increasing the membrane charge resulted in dewetting, for this system it results in complete wetting. This is presumably due to the net positive charge of PDDA/ATP droplets¹⁵. We also tested condensates made of oligopeptides of lysine (K₁₀) and aspartic acid (D₁₀). These amino acids are common in repeating sequences of phase-separating proteins in cells, and have been used as minimal models for key types of interactions leading to phase separation and interaction with RNA¹⁸. In this case, the inclusion of a negative charge on the membrane promoted complete wetting, however, to achieve complete dewetting it was necessary to increase the ionic strength of the medium (see below new Figure S9).

Figure S9. Wetting transitions for pure DOPC or DOPC:DOPS GUVs in contact with poly-lysine (K_{10}) and poly-aspartic acid (D_{10}) droplets. (a) Chemical structures of the oligopeptides poly-lysine (K_{10}) and poly-aspartic acid (D_{10}). Each peptide contains 10 monomeric units ($n=10$, as indicated). (b) Examples of confocal fluorescence and bright-field images of K_{10}/D_{10} condensates labeled with TAMRA- K_{10} . (c) Confocal microscopy images of vesicles (magenta) in contact with K_{10}/D_{10} droplets (yellow). The first two images are for pure DOPC vesicles in the presence of 20 mM NaCl (left) or 0 mM NaCl (right), and the last two images correspond to a binary mixture of DOPC with the indicated % mol of DOPS in the absence of NaCl. To visualize two wetting transitions in this system it is necessary to change two parameters, namely the membrane charge to achieve complete wetting, and the salinity of the medium to achieve complete dewetting. Scale bars: 5 μm .

These new results show that it is possible to observe the complete dewetting/partial wetting/complete wetting transitions for chemically very different systems. Nevertheless, it is important to reiterate that for each particular system the conditions under which these transitions can take place may differ. Wetting transitions can be tuned by membrane composition, condensate composition, and salinity as shown, but other factors such as pH and temperature could also influence the wetting morphologies. We added the following text on page 13:

“The above results on membrane remodeling and wetting transitions were obtained using glycinin droplets, but biomolecular condensates can exhibit a wide range of surface charge, interfacial tension and viscosity⁸. It is important to note that the wetting state can be expected to depend not only on the material properties of the condensate, but should be also molded by the specific interactions between particular lipids (or membrane receptors) and the condensate components. While for glycinin droplets the inclusion of charges in the membrane promotes dewetting, for other condensates the presence of charges could favour wetting. This is exemplified by Dengue and Zika virus capsid proteins, which form condensates with DNA/RNA and wet negatively charged membranes¹³. To demonstrate this, we performed experiments showing wetting transitions with

condensate droplets made of a polyamine and adenosine triphosphate (ATP), as well as from the oligopeptide pair poly-lysine (K_{10}) and poly-aspartic acid (D_{10}), see Figs. S8 and S9. All three regimes (complete and partial wetting and dewetting) can be observed suggesting a universal pattern. The transitions in these highly charged systems (positive for PDDA/ATP droplets¹⁵ and negative for K_{10}/D_{10} condensates) can also be modulated by membrane composition and salinity as demonstrated for glycinin, albeit at different conditions (compare Fig. 3 and Figs. S8 and S8). These results point to a widespread behavior and show that depending on the chemical nature of the condensates, different parameters should be tuned to produce wetting transitions.”

Finally, as requested by the Reviewer, we have now included data on the changes in the geometric factor for the PEG/Dextran system, see new panel Fig. S1b. Here, the independent variable is the deflation ratio (i.e. polymer concentration) rather than sodium chloride concentration (or membrane composition) as in the case of glycinin. In this system, only dewetting to partial wetting transition is accessible experimentally^{14, 19}.

5) Would the force distribution follow similar patterns if the size of the GUVs is smaller than the condensates?

The force balance between the interfacial tensions shown in Fig. 1c does not depend on the size of the vesicle/condensate pair. The factors that can alter the balance are the composition of the external milieu, the composition of the membrane and the chemical nature of the condensates. We now demonstrate this by including an example in new Fig. S11 showing that pairs with different condensate and vesicle sizes have the same geometric factor at a given condition:

Figure S11. Changes in the vesicle and condensate sizes do not alter the geometric factor calculation. The interfacial tensions force balance shown in Fig. 2c is not altered by the respective sizes of the GUV and condensate interacting. The images show two examples of DOPC vesicles in contact with K_{10}/D_{10} droplets. In image 1, the droplet is bigger than the vesicle and the opposite case is shown in image 2. The respective sizes of the droplets and vesicles are plotted in the graph. Despite the size differences, the force balance does not change, and the angles formed between the droplet and the membrane are the same, giving in both cases a geometric factor of $\Phi \approx 0.6$. Scale bars: 5 μm .

6) it would be nice to have theoretical model to strengthen their conclusions. But I understand this will be a tremendous effort and thus it is authors' choice.

We have indeed developed a theoretical model that supports our findings. The geometric factor plotted in Fig 1, comes from this model, which is introduced and now expanded in the Methods section “From

contact angles to fluid-elastic parameters” starting on page 22. The model allows quantifying the different wetting morphologies. From the Reviewer’s comment, we realized the description was too short and not clearly presented in the previous version of the manuscript. The theoretical model was included and expanded in the Methods section. We have now strengthened it and presented it in a more visible manner. We also added a new theoretical section describing the condition for ruffling – new Methods section:

“Criterion for ruffling of the condensate-membrane interface. Ruffling and fingering of the ic interface implies an increase of the bending energy (assuming negligible spontaneous curvature). Ruffling will be observed when this bending energy is overcompensated by the gain in adhesion energy for transferring membrane area from the ie (condensate-free) segment to the ic interface (in contact with the condensate). When we transfer the membrane area ΔA , the gain in adhesion energy is $E_{ad} = W\Delta A$. Introducing Eq. (11) yields $E_{ad} = \Phi\Sigma_{ce}\Delta A$, which is negative for negative Φ ; note that ruffling is observed only at for $[NaCl] \geq 100$ mM where $\Phi < 0$, see Fig. 1. The bending energy E_{be} is proportional to $8\pi\kappa$ and also proportional to the number N_{pro} of protrusions formed by the membrane area ΔA . The total energy change associated with the transferred membrane area is then given by $\Delta E = E_{ad} + E_{be} = \Phi\Sigma_{ce}\Delta A + c8\pi\kappa N_{pro}$. Here $\Phi < 0$ and c is a dimensionless coefficient of the order of 1. This energy change ΔE is negative for $|\Phi|\Sigma_{ce}\Delta A > c8\pi\kappa N_{pro}$ (where $|\Phi|$ is the absolute value of Φ). Ignoring the dimensionless coefficient c , we obtain the simple criterion

$$|\Phi|\Sigma_{ce}\Delta A > 8\pi\kappa N_{pro} \quad (12)$$

for the area transfer and ruffling to be energetically favorable. To examine this relation, we consider the vesicle condensate pair shown in Fig. 8b. For the interfacial tension and the bending rigidity we take $\Sigma_{ce} \simeq 0.5$ mN/m (as measured here) and $\kappa \simeq 10^{-19}$ J. The area change stored in ruffles that can be pulled out with a micropipette is of the order of $50 \mu\text{m}^2$. For the salinity conditions where ruffling is observed ($[NaCl] \geq 100$ mM), $|\Phi|$ ranges between 0.2 and 1, see Fig. 1. If we take for the number of protrusions $N_{pro} = 100$, which is a generous overestimate, we still obtain that the criterion Eq. (12) is satisfied whereby the adhesion energy gain is still orders of magnitude higher than the bending energy penalty for ruffling.

Judging from the images in Figs. 6-8, the fingers and undulations do not have characteristic length scale. Their dimensions are defined by the available membrane area (ΔA) to form the protrusions and the volume constraints imposed by the condensate and vesicle sizes.”

Minor Concerns:

1) The manuscript has some typos needed to be corrected.

The manuscript was revised and the typos we found were corrected.

2) In figure 5, what determines whether the membrane condensate mixture would be involved in bridging or would undergo an active engulfment procedure?

We thank the Reviewer for this question. The answer to bridging is somewhat trivial. Bridging of membranes by condensates will occur provided the system is in the state of partial wetting, and the membranes are close together and able to interact with the same condensate. For engulfment to occur, the droplet size has to be smaller or equal to the excess area of the vesicle (compared to the area of a sphere with the vesicle volume) available for deformation. Again, partial or complete wetting is a requirement. This is specified on page 8:

“The necessary condition for engulfment is partial or complete wetting and excess vesicle area (compared to the area of a sphere enclosing the same volume) sufficient to wrap the condensate.”

3) What are the parameters and thresholds used to consider the separate phases of dewetting, partial and complete wetting?

The different wetting regimes are evaluated according to the geometric factor plotted in Fig. 2 and defined in Methods, Eq. (8-10). The geometric factor is calculated from the angles that the droplet membrane system forms (see Fig 1b). It is equal to the rescaled affinity contrast, W/Σ_{ce} , see Eq. 11 in the Methods, which has a direct mechanical interpretation in terms of the (de)wetting behavior. The largest possible value of the rescaled affinity contrast is equal to $W/\Sigma_{ce} \equiv \Phi = 1$, which corresponds to complete dewetting from the condensate droplet, the smallest value given by -1 to complete wetting by the condensate droplet, and the intermediate values describe partial dewetting and partial wetting morphologies.

4) All images in a figure panel should have the same scale for a better understanding of the readers. This needs to be corrected for all figures. The addition of scale bar in all the images is needed to make the figures consistent.

We thank the Reviewer for noticing this. All images have scale bars now and their value is indicated in the figure captions.

5) It would be good if the schematics of the experimental workflow are added to the figures before representing the results.

The experimental workflow consists of mixing the vesicles with the condensates and looking for occasions where the two are in contact. The only difference could be the incubation time which is longer for the study on wetting/ruffling dynamics which is specified in the caption. The arrows in the schematics of Fig. 1 do not really represent the workflow but rather the imposed changes in conditions before mixing (salinity, charge).

6) In figure 3b, the number of experiments is not mentioned for SD calculations.

We have included it now in the figure legend.

Reviewer #2 (Remarks to the Author):

This manuscript uses a model in vitro system comprised of biomolecular condensates formed by the protein glycinin, together with membrane vesicles, to examine the biophysical basis of condensate-membrane wetting behavior. The authors show that there is a salt concentration dependent changes in this wetting behavior, which can be tuned from completely non-wetting to completely wetting. They report and examine a number of interesting features, including a surprising slow relaxation to equilibrium contact angles, as well as a differential mobility of membranes on the different interfaces (the condensate wetting appears to significantly slow membrane mobility). Probably most strikingly, the authors describe a fascinating behavior in which the membrane ruffles, and use micropipette aspiration experiments to show how this appears to be related to excess membrane availability and associated membrane tension. Overall, I find this to be a very solid manuscript, that examines a timely and interesting problem, and thus

I think it is quite suitable for publication in Nature Communications. I have only relatively minor suggestions that can be incorporated in the revised manuscript.

We thank the Reviewer for appreciating the novelty, quality, and timeliness of our work and for the comments and suggestions that have helped us improve the manuscript.

Minor comments:

1. The authors refer to the data as describing “three different wetting regimes separated by two wetting transitions”. Is it really appropriate to refer to these as different wetting regimes, when ultimately this is behavior that can be described by a single continuous variable (ϕ)?

The wetting regimes are described by the rescaled affinity contrast W/Σ_{ce} which turns out to be equal to the geometric factor Φ . The three distinct cases are $W/\Sigma_{ce} \equiv \Phi = 1$ which describes complete dewetting, $W/\Sigma_{ce} \equiv \Phi = -1$ corresponding to complete wetting and the intermediate values correspond to partial dewetting and wetting morphologies. In the context of wetting phenomena at solid substrate surfaces, it is well understood that partial wetting and complete wetting represent two different wetting regimes. Likewise, dewetting and partial wetting correspond to two different wetting regimes as well.

2. The authors describe the membrane kink as reflecting $\sqrt{\kappa/\sigma_{ce}}$, but don't use the term “Elastocapillary length”. Is there a reason for this? I would suggest calling it that, is it is a pretty well-known name for that quantity.

We thank the Reviewer for this interesting comment. It is important to note, that the elastocapillary length is quite different from our curvature radius $\sqrt{\kappa/\Sigma_{ce}}$ as one can immediately conclude from the dependence of these two length scales on the interfacial tension: our curvature radius decreases with increasing interfacial tension whereas the elastocapillary length as discussed in the literature (new references Style et al. and Wang et al.) increases with increasing interfacial tension. This difference becomes obvious from the different physical dimensions of bending rigidity κ and Young's modulus Y . Whereas the bending energy κ has the units of energy, the Young's modulus Y has the units of energy per volume. We have now included the following text on page 7:

“In the context of liquid droplets at the surfaces of solid materials, the competition between the surface tension Σ_{int} of the liquid-vapor interface and the Young's modulus Y of the solid material leads to the elastocapillary length Σ_{int}/Y ^{20, 21}. Note that the elastocapillary length grows with increasing surface tension whereas the curvature radius $\sqrt{\kappa/\Sigma_{ce}}$ decreases with increasing interfacial tension.”

3. In the methods, it says “..and the viscosity, close to that of the nucleus”; this should be “nucleolus”.

We corrected the typo.

4. I found the description of the physics underlying the ruffling behavior somewhat unclear/unsatisfying. Perhaps the authors could say a bit more, in particular about what sets the length scale of ruffling. This is presumably also related to the elastocapillary length, although the curvatures here appear to be smaller than the high curvature of the “kink” regions at the contact lines.

We now address the criticism of the Reviewer regarding the underlying physics by including an additional explanation on page 11:

“The intricate and striking membrane morphologies arise from the gain of adhesion energy between droplet and membrane when excess area is pulled out of the *ie* segment and added to the *ic* segment, see Methods. Subsequently, the tension of this segment becomes negligible $\Sigma_{ic}^m \approx 0$, which implies, via the force balance triangle (Fig. 2c) that $\Sigma_{ic}^m / \Sigma_{ce} \equiv \sin \theta_e / \sin \theta_i \approx 0$ and that the angle θ_e should approach 180° . This is indeed consistent with our experimental observations that show that the *ic* membrane segment becomes parallel to the *ce* interface along the contact line, i.e. the vesicle-condensate pair rounds up overall (see e.g. Figs. 6a, 7c). Furthermore, this tension relation together with Eqs. (10) and (11) imply negative values for the geometric factor Φ , and indeed, interface ruffling was observed only for $[\text{NaCl}] \geq 100 \text{ mM}$ where $\Phi < 0$, i.e. for negative affinity contrast.”

We thank the Reviewer for the question about the length scale of ruffling. It does not seem to be set by the material properties of the system as we observe various sizes of the protrusions for the same system (compare data in Figs. 6-8). We believe that the sizes of the fingers and ruffles is set by the available membrane area of the *ic* segment (which provides the contact area with the condensate) and volume constraints imposed by the vesicle and condensate sizes limiting the depth which the protrusions can explore. This is now summarized on page 24 as follows:

“Judging from the images in Figs. 6-8, the fingers and undulations do not have characteristic length scale. Their dimensions are defined by the available membrane area (ΔA) to form the protrusions and the volume constraints imposed by the condensate and vesicle sizes.”

References:

1. Herman EM, Larkins BA. Protein Storage Bodies and Vacuoles. *The Plant Cell* **11**, 601-613 (1999).
2. Feeney M, Kittelmann M, Menassa R, Hawes C, Frigerio L. Protein Storage Vacuoles Originate from Remodeled Preexisting Vacuoles in *Arabidopsis thaliana*. *Plant Physiology* **177**, 241-254 (2018).
3. Yoo BY, Chrispeels MJ. The origin of protein bodies in developing soybean cotyledons: a proposal. *Protoplasma* **103**, 201-204 (1980).
4. Baranova EN, Gulevich AA, Kalinina-Turner EB, Koslov NN. Effects of NaCl, Na₂SO₄ and mannitol on utilization of storage protein and transformation of vacuoles in the cotyledons and seedling roots of alfalfa (*Medicago sativa* L.). *Russian Agricultural Sciences* **37**, 11-19 (2011).
5. Pombo-García K, Martin-Lemaitre C, Honigmann A. Wetting of junctional condensates along the apical interface promotes tight junction belt formation. *bioRxiv*, 2022.2012.2016.520750 (2022).
6. Sezgin E, Kaiser HJ, Baumgart T, Schwille P, Simons K, Levental I. Elucidating membrane structure and protein behavior using giant plasma membrane vesicles. *Nat Protoc* **7**, 1042-1051 (2012).
7. Chen N, Zhao Z, Wang Y, Dimova R. Resolving the Mechanisms of Soy Glycinin Self-Coacervation and Hollow-Condensate Formation. *ACS Macro Letters* **9**, 1844-1852 (2020).
8. Wang H, Kelley FM, Milovanovic D, Schuster BS, Shi Z. Surface tension and viscosity of protein condensates quantified by micropipette aspiration. *Biophysical Reports* **1**, 100011 (2021).
9. Yuan F, et al. Membrane bending by protein phase separation. *Proceedings of the National Academy of Sciences* **118**, e2017435118 (2021).
10. Babl L, Merino-Salomón A, Kanwa N, Schwille P. Membrane mediated phase separation of the bacterial nucleoid occlusion protein Noc. *Scientific Reports* **12**, 17949 (2022).
11. Wang H-Y, et al. Coupling of protein condensates to ordered lipid domains determines functional membrane organization. *bioRxiv*, 2022.2008.2002.502487 (2022).

12. Wan-Chih Su DG, Andrew Rowland, Christine Keating, Atul Parikh. Liquid-Liquid Phase Separation inside Giant Vesicles Drives Shape Deformations and Induces Lipid Membrane Phase Separation. *Research Square*, (2022).
13. Ambroggio EE, Costa Navarro GS, Pérez Socas LB, Bagatolli LA, Gamarnik AV. Dengue and Zika virus capsid proteins bind to membranes and self-assemble into liquid droplets with nucleic acids. *Journal of Biological Chemistry* **297**, 101059 (2021).
14. Zhao Z, Roy D, Steinkühler J, Robinson T, Lipowsky R, Dimova R. Super-resolution imaging of highly curved membrane structures in giant vesicles encapsulating molecular condensates. *Advanced Materials* **34**, 2106633 (2021).
15. Lu T, *et al.* Endocytosis of Coacervates into Liposomes. *Journal of the American Chemical Society* **144**, 13451-13455 (2022).
16. Dora Tang TY, *et al.* Fatty acid membrane assembly on coacervate microdroplets as a step towards a hybrid protocell model. *Nature Chemistry* **6**, 527-533 (2014).
17. Poudyal RR, Pir Cakmak F, Keating CD, Bevilacqua PC. Physical Principles and Extant Biology Reveal Roles for RNA-Containing Membraneless Compartments in Origins of Life Chemistry. *Biochemistry* **57**, 2509-2519 (2018).
18. Choi S, Meyer MO, Bevilacqua PC, Keating CD. Phase-specific RNA accumulation and duplex thermodynamics in multiphase coacervate models for membraneless organelles. *Nature Chemistry*, (2022).
19. Li Y, Lipowsky R, Dimova R. Transition from complete to partial wetting within membrane compartments. *Journal of the American Chemical Society* **130**, 12252-12253 (2008).
20. Chen L, Bonaccorso E, Gambaryan-Roisman T, Starov V, Koursari N, Zhao Y. Static and dynamic wetting of soft substrates. *Current Opinion in Colloid & Interface Science* **36**, 46-57 (2018).
21. Style RW, Boltanskiy R, Che Y, Wettlaufer JS, Wilen LA, Dufresne ER. Universal Deformation of Soft Substrates Near a Contact Line and the Direct Measurement of Solid Surface Stresses. *Physical Review Letters* **110**, 066103 (2013).

REVIEWERS' COMMENTS

Reviewer #1 (Remarks to the Author):

The authors have addressed most of my questions. One of my major concerns were about the how widely can the theory be applied given that the initial manuscript contained observations made from very specific coacervate-membrane interactions. However, the authors now have shown with multiple examples, to our satisfaction, that their model indeed applies to wider range of coacervate and membrane complexes. The authors have also elaborated on the mathematical model more clearly. They have also shown the relevance of coacervate-mediated membrane ruffling and fingering events in the case of glycinin -tonoplast membrane systems in soybean cotyledon cells and in Arabidopsis cells, which establishes their point in the cell biological context.

I have only a couple of minor concerns.

1. the cellular relevance part i.e. the glycinin-tonoplast membrane images of figure 9 were obtained from other publications. The authors said that they obtained them with permission. Even though this part adds value to the manuscript, I am not sure if this should be a main figure in the main text, perhaps in the supplementary file.
2. The manuscript has many typos, grammatical and punctuation errors. but this is more of an editorial concern than a scientific one.

Reviewer #2 (Remarks to the Author):

The authors have done a thorough job of addressing the comments I made on the initially submitted version of the manuscript, and I believe the manuscript is ready for publication.

One additional remark: In one of my comments to the reviewers, I mentioned that the authors might want to use the term "elasto-capillary length", for $\text{length} = \sqrt{\kappa / \text{surface tension}}$. In the authors' response, they make a distinction between this quantity, which they refer to as the "curvature radius", and what they say is actually the elastocapillary length, defined as $\text{length} = \text{surface tension} / \text{young's modulus}$.

The authors point out that these are two very different quantities, with one decreasing and the other increasing with greater surface tension. The authors point to references that use the $\text{length} = \text{surface tension} / \text{young's modulus}$ as a definition of elastocapillary length, including in the following article: <https://doi.org/10.1103/PhysRevLett.110.066103>

However, it seems that there is some ambiguity in the literature about what should be called the elastocapillary length. For example, see <https://iopscience.iop.org/article/10.1088/0953-8984/22/49/493101/pdf>, or <https://doi.org/10.1038/s41586-022-05138-6>

ANSWERS TO REVIEWERS' COMMENTS

(Reviewers' comments in black; answers in blue)

Reviewer #1:

The authors have addressed most of my questions. One of my major concerns were about the how widely can the theory be applied given that the initial manuscript contained observations made from very specific coacervate-membrane interactions. However, the authors now have shown with multiple examples, to our satisfaction, that their model indeed applies to wider range of coacervate and membrane complexes. The authors have also elaborated on the mathematical model more clearly. They have also shown the relevance of coacervate-mediated membrane ruffling and fingering events in the case of glycinin - tonoplast membrane systems in soybean cotyledon cells and in Arabidopsis cells, which establishes their point in the cell biological context.

I have only a couple of minor concerns.

1. the cellular relevance part i.e. the glycinin-tonoplast membrane images of figure 9 were obtained from other publications. The authors said that they obtained them with permission. Even though this part adds value to the manuscript, I am not sure if this should be a main figure in the main text, perhaps in the supplementary file.

We find this figure important for the following reason. It shows in vivo evidence for membrane deformation by condensates, which strengthens the biological relevance of the work (which was also requested by the Reviewer). We would like to draw attention to the bulk of studies reported in the last couple of decades which show condensates examined with electron microscopy. Only now these studies can be considered from the fresh perspective and knowledge about condensate fluidity.

Furthermore, before submitting the last revision, we acquired the permissions to reproduce.

2. The manuscript has many typos, grammatical and punctuation errors. but this is more of an editorial concern than a scientific one.

We thank the Reviewer for the comment. We have now had the manuscript proofread by a native speaker.

Reviewer #2:

The authors have done a thorough job of addressing the comments I made on the initially submitted version of the manuscript, and I believe the manuscript is ready for publication.

One additional remark: In one of my comments to the reviewers, I mentioned that the authors might want to use the term "elasto-capillary length", for $\text{length} = \sqrt{\kappa/\text{surface tension}}$. In the authors response, they make a distinction between this quantity, which they refer to as the "curvature radius", and what they say is actually the elastocapillary length, defined as $\text{length} = \text{surface tension}/\text{young's modulus}$.

The authors point out that these are two very different quantities, with one decreasing and the other increasing with greater surface tension. The authors point to references that use the $\text{length} = \text{surface tension}/\text{young's modulus}$ as a definition of elastocapillary length, including in the following article: <https://doi.org/10.1103/PhysRevLett.110.066103>

However, it seems that there is some ambiguity in the literature about what should be called the elastocapillary length. For example, see <https://iopscience.iop.org/article/10.1088/0953-8984/22/49/493101/pdf>, or <https://doi.org/10.1038/s41586-022-05138-6>

We thank the reviewer for the comments and for recommending our work for publication. Regarding the elastocapillary length, we agree with the Reviewer. For droplets wetting thin solid plates, another definition of the elastocapillary length has been used in the literature and this alternative definition is analogous to the length scale that we use. To clarify this issue, we have now added another sentence to the paragraph in which we discuss the elastocapillary length, and have included the reference provided by the reviewer:

“For droplets at thin solid plates, the competition between the bending modulus B and the interfacial tension Σ_{int} leads to the length scale $\sqrt{B/\Sigma_{ce}}$ which provides another definition of an “elastocapillary length”¹. The latter definition is analogous to the length scale $\sqrt{\kappa/\Sigma_{ce}}$ considered here in the context of fluid membranes.”

1. Bico J, Reyssat É, Roman B. Elastocapillarity: When Surface Tension Deforms Elastic Solids. *Annual Review of Fluid Mechanics* **50**, 629-659 (2018).